# Less is More: Masking Elements in Image Condition Features Avoids Content Leakages in Style Transfer Diffusion Models

**Lin Zhu[1], Xinbing Wang[1], Chenghu Zhou[2], Qinying Gu[3], Nanyang Ye[1]** *

[1] Shanghai Jiao Tong University, [2] Chinese Academy of Sciences,
[3] Shanghai Artificial Intelligence Laboratory
{zhulin_sjtu, xwang8, ynylincoln}@sjtu.edu.cn,
zhouchsjtu@gmail.com, guqinying@pjlab.org.cn

## Abstract

Given a style-reference image as the additional image condition, text-to-image diffusion models have demonstrated impressive capabilities in generating images that possess the content of text prompts while adopting the visual style of the reference image. However, current state-of-the-art methods often struggle to disentangle content and style from style-reference images, leading to issues such as *content leakages*. To address this issue, we propose a masking-based method that efficiently decouples content from style without the need of tuning any model parameters. By simply masking specific elements in the style reference's image features, we uncover a critical yet under-explored principle: guiding with appropriately-selected fewer conditions (e.g., dropping several image feature elements) can efficiently avoid unwanted content flowing into the diffusion models, enhancing the style transfer performances of text-to-image diffusion models. In this paper, we validate this finding both theoretically and experimentally. Extensive experiments across various styles demonstrate the effectiveness of our masking-based method and support our theoretical results.

## 1 Introduction

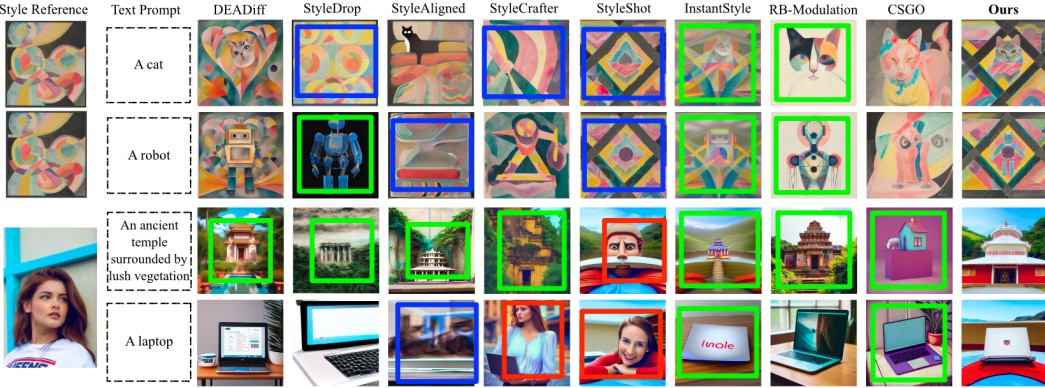

Figure 1: Given a style-reference image, our method is capable of synthesizing new images that resemble the style and are faithful to text prompts simultaneously. Previous methods often face issues of either content leakages or style degradation. We mark the results with significant content leakages, style degradation, and loss of text fidelity with red, green, and blue boxes, respectively.

---

*Nanyang Ye is the corresponding author. The code is available at `https://github.com/LinLLLL/MaskST`.

Recently, text-to-image diffusion models (Zhao et al., 2024b; Saharia et al., 2022; Zhang et al., 2023a; Zhu et al., 2023; Zhang et al., 2023b) have achieved notable success in generating high-quality images, especially for tasks requiring personalized image creation that preserves specific stylistic elements. By incorporating a style-reference image as an additional input, recent approaches (Park et al., 2024; Hamazaspyan & Navasardyan, 2023; Chung et al., 2024; Wang et al., 2023b; Qi et al., 2024; Zhang et al., 2023c) have effectively synthesized images that not only align with the content described in text prompts but also adopt the visual style of the reference image. However, despite these advancements, content leakages from style-reference images remain a persistent issue (Wang et al., 2024a; Ruiz et al., 2023; Jeong et al., 2024), as illustrated in Figure 1. *Content leakages* occurs when intensifying the style transfer causes unintended non-stylistic elements from the reference image to be incorporated into the generated output (Wang et al., 2024a). Conversely, reducing style intensity to prevent content leakage can result in *style degradation*, hindering effective style transfer (Jeong et al., 2024). These challenges highlight the difficulty of disentangling styles from contents in style-reference images.

Some approaches (Zhang et al., 2018a;b; Qi et al., 2024) try to achieve the disentanglement by constructing paired datasets in which images share the same subject but exhibit distinct styles, facilitating the extraction of disentangled style and content representations. Other works (Sohn et al., 2024; Liu et al., 2023; Zhang et al., 2024) optimize some or all of the model parameters using large sets of diverse style images, allowing them to isolate and integrate these stylistic elements into diffusion models. However, due to the inherently ambiguous nature of style, building comprehensive style datasets is resource-intensive and limits the model's capacity to generalize to styles not present in the dataset. To address this issue, InstantStyle (Wang et al., 2024a) proposed a training-free strategy to separate style from content by subtracting content-related features from image features. Although this approach is simple and training-free, feature subtraction across different modalities inevitably introduces the image-text misalignment issue (Kim et al., 2023; Gordon et al., 2023), which hinders accurate disentanglement of content and style. As illustrated in Figure 1, although InstantStyle mitigates content leakage, it comes at the cost of significant style degradation.

To overcome all these limitations, we propose a simple and effective training-free method that efficiently decouples content from style, without requiring tuning any model parameters. Unlike InstantStyle, which subtracts features across different modalities, our approach removes content from the style-reference image by masking the image feature elements associated with the content. Specifically, we identify these content-related elements through clustering the element-wise product of the style-reference image features and the content text features, and then set their values to zero. The theoretical evidence for the effectiveness of this identification approach is presented in Proposition 1.

By simply masking specific elements in the style reference's image features, we uncover a critical yet under-explored principle: guiding with appropriate masked conditions (e.g., masking several image feature elements) can prevent undesired content information from leaking into diffusion models, thereby improving style transfer performance. We further present theoretical evidence for this principle. As demonstrated in Theorem 1 and Theorem 2, diffusion models guided by fewer appropriately selected conditions (e.g., the masked image feature and the text feature) achieve a lower divergence between the generated and real image distributions compared to models relying on more conditions that are less coherent (e.g., unfiltered image features combined with text and additional content features). This result aligns with the concept that "Less is more". Extensive experiments across various styles, along with comparisons to state-of-the-art methods, validate the effectiveness of our approach and support our theoretical findings.

## 2 PRELIMINARIES

Because of the portability and efficiency, we illustrate the proposed masking-based method based on the baseline module IP-Adapter (Ye et al., 2023). In this section, we present the background knowledge and key observations from our initial experiments as follows:

**Conditional Diffusion Models** Diffusion models consist of two processes: a diffusion process (forward process), which incrementally adds Gaussian noise $\epsilon$ to the data $x_0$ through a Markov chain. Additionally, a denoising process generates samples from Gaussian noise $x_T \sim N(0, 1)$ with a learnable denoising model $\hat{\epsilon}_\theta(x_t, t, c)$ parameterized by $\theta$, This denoising model $\epsilon_\theta(\cdot)$ is imple-

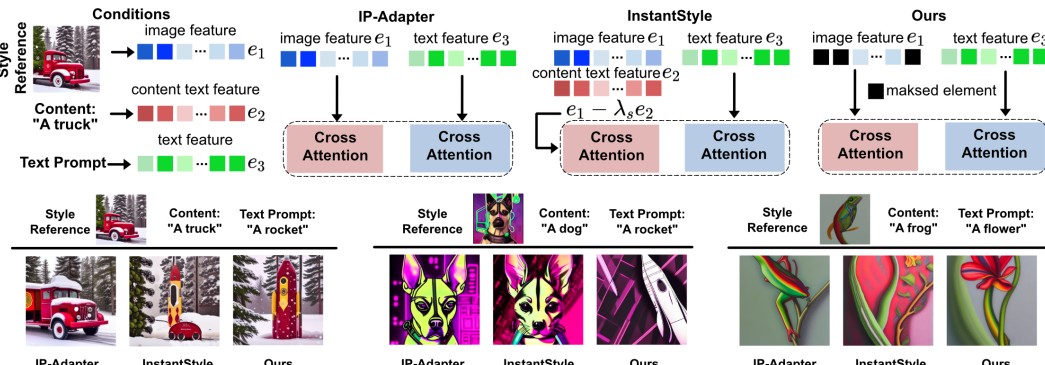

Figure 2: **Top:** The differences in the conditions between IP-Adapter (Ye et al., 2023), InstantStyle (Wang et al., 2024a), and Ours. We elaborate on how to select masked elements in Section 3.1. **Bottom:** Illustration of the content leakages issue.

mented with U-Net and trained with a mean-squared loss derived by a simplified variant of the variational bound: $\mathcal{L} = \mathbb{E}_{t,\boldsymbol{x}_0,\boldsymbol{\epsilon}} \left[ \|\boldsymbol{\epsilon} - \hat{\boldsymbol{\epsilon}}_\theta(\mathbf{x}_t, t, \boldsymbol{c})\|^2 \right]$, where $\boldsymbol{x}_0$ represents the real data with an additional condition $\boldsymbol{c}$, $t \in [0, T]$ denotes the time step of diffusion process, $\boldsymbol{x}_t = \alpha_t \boldsymbol{x}_{t-1} + \sigma_t \boldsymbol{\epsilon}$ is the noisy data at $t$ step, and $\alpha_t$, $\sigma_t$ are predefined functions of $t$ that determine the diffusion process. For conditional learning, classifier-free guidance (Ho & Salimans, 2022) is often employed, which uses a single neural network to parameterize both the conditional model and unconditional model, where for the unconditional model one can simply input a null token $\varnothing$ for the text features $c$ when predicting the noise, i.e. $\boldsymbol{\epsilon}_\theta(\boldsymbol{x}_t, t) = \boldsymbol{\epsilon}_\theta(\boldsymbol{x}_t, t, \boldsymbol{c} = \varnothing)$. Then one can perform sampling using the following linear combination of the conditional and unconditional noise estimates: $\tilde{\boldsymbol{\epsilon}}_\theta(\boldsymbol{x}_t, t, \boldsymbol{c}) = \omega \boldsymbol{\epsilon}_\theta(\boldsymbol{x}_t, t, \boldsymbol{c}) + (1 - \omega) \boldsymbol{\epsilon}_\theta(\boldsymbol{x}_t, t)$. Once the model $\boldsymbol{\epsilon}_\theta(\cdot)$ is trained, images can be generated from random noises in an iterative manner.

**IP-Adapter** As "an image is worth a thousand words", IP-Adapter (Ye et al., 2023) proposed an effective and lightweight adapter to achieve image prompt capability for the pre-trained text-to-image diffusion models. It uses two decoupled cross-attention modules to process text and image conditions and finally performs linear weighting. Given the query features $\mathbf{Z}$, the text features $\boldsymbol{c}_t$, and the image features $\boldsymbol{c}_i$, the final formulation of the two cross-attention modules is defined as:

$$\mathbf{Z}^{new} = \text{Softmax}(\frac{\mathbf{Q}\mathbf{K}^\top}{\sqrt{d}})\mathbf{V} + \text{Softmax}(\frac{\mathbf{Q}(\mathbf{K}')^\top}{\sqrt{d}})\mathbf{V}'$$
$$\text{where } \mathbf{Q} = \mathbf{Z}\mathbf{W}_q, \mathbf{K} = \boldsymbol{c}_t\mathbf{W}_k, \mathbf{V} = \boldsymbol{c}_t\mathbf{W}_v, \mathbf{K}' = \boldsymbol{c}_i\mathbf{W}'_k, \mathbf{V}' = \boldsymbol{c}_i\mathbf{W}'_v \qquad (1)$$

where $\mathbf{Q}$, $\mathbf{K}$ and $\mathbf{V}$ are the query, key, and values matrices from the text features; $\mathbf{K}'$ and $\mathbf{V}'$ are the key, and values matrices from the image features. IP-Adapter used the same query for image cross-attention as for text cross-attention. The weight matrices $\mathbf{W}_q$, $\mathbf{W}_k$, and $\mathbf{W}_v$ correspond to text cross-attention and remain frozen, consistent with the original pre-trained model. Only the weight matrices in image cross-attention, $\mathbf{W}'_k$ and $\mathbf{W}'_v$, are trainable. In the inference stage, one can also adjust the weight of the image condition:

$$\mathbf{Z}^{new} = \text{Attention}(\mathbf{Q}, \mathbf{K}, \mathbf{V}) + \lambda_i \cdot \text{Attention}(\mathbf{Q}, \mathbf{K}', \mathbf{V}') \qquad (2)$$

where $\lambda_i$ is the coefficient of image conditions, and the model becomes the original text-to-image diffusion model if $\lambda_i = 0$.

**InstantStyle and Limitations** Fully compatible with IP-Adapter, InstantStyle (Wang et al., 2024a) employs block-specific injection techniques to achieve style transfer. Additionally, it proposes an efficient method to decouple content and style from style references, highlighting that straightforward subtraction of content text features from image features can effectively reduce content leakages. However, this approach has some limitations: 1) On the one hand, the feature subtraction is based on CLIP's embeddings, and it relies on the assumption that CLIP global features provide a robust characterization for explicit decoupling. It means that the process necessitates good alignment between CLIP's image-text features for all style references, which may be unrealistic for complex style

references. 2) On the other hand, tuning the coefficient of image condition (i.e., $\lambda_i$ in Equation 2) is important for its effect in addressing content leakages, but it is labour-intensive and very tricky [1].

To comprehensively understand the content leakages issue, we present visualization examples for IP-Adapter and InstantStyle in the context of style transfer. Given a style-reference image as the additional image condition, the models are tasked with generating images that reflect the text prompt while incorporating the style of the image reference. Based on Stable Diffusion V1.5 (SD1.5, (Rombach et al., 2022)), we adopt DDIM sampler (Song et al., 2022) with 30 steps, set the guidance scale to 7.5, and the coefficient of image condition to 1.0. As illustrated in Figure 2, IP-Adapter struggles with maintaining the presence of objects in text prompts. Due to image-text misalignment, InstantStyle also fails to achieve seamless synthesis of text prompts and reference styles.

**Effectiveness of Less Condition** To mitigate image-text misalignment, we focus on feature manipulation within the image feature space instead of introducing feature subtraction between different modalities. Specifically, we propose to eliminate content from the style-reference image by discarding the image feature elements associated with that content. To achieve this, we mask the content-related elements by setting their values to zero. The content-related elements are identified through clustering the element-wise product of the style reference image features and the content text features. As shown in Figure 2, our method successfully achieves more accurate style transfer by masking certain elements in the style reference. Moreover, as illustrated in Figure 10, compared to InstantStyle, our method can achieve more stable results across various coefficient values of $\lambda_i$, particularly in high-coefficient scenarios. Thus, we arrive at the key motivation of this paper:

*The experiment results in Figure 1 and Figure 2 suggest that leveraging appropriately-selected fewer conditions, such as the masked image features, surprisingly effectively avoids content leakages, thereby enhancing text-to-image models in style transfer.*

Motivated by this observation, we theoretically and experimentally explore the role of the masking strategy in eliminating content leakages in text-to-image diffusion models. In this paper, we demonstrate that our masking-based method can outperform recent state-of-the-art methods without the need for tuning model parameters or the coefficient value of $\lambda_i$, which we fix at 1.0.

## 3    METHODOLOGY

In Section 3.1, we elaborate on our novel masking-based method for efficiently decoupling content from style. This approach utilizes a masking strategy for image features, where the masked image feature elements are identified through clustering on the element-wise product of image features and content text features. We also provide supporting evidence for the effectiveness of this masked element selection method in Proposition 1. Furthermore, in section 3.2, we theoretically demonstrate that our method surpasses InstantStyle's feature subtraction by achieving a smaller divergence between the generated image distribution and the real image distribution (See Theorem 1). To delve deeper, we also investigate whether the effectiveness of appropriately fewer conditions holds in tuning-based models. We present the theoretical results in Theorem 2.

### 3.1    THE PROPOSED MASKING-BASED METHOD FOR DECOUPLING CONTENT FORM STYLE

Before delving into the details of our method, we first present the important notations as follows:

**Notations 1** *Let $q(\boldsymbol{x}|c)$ be the joint distribution for the data $\boldsymbol{x}$ and the condition $c$ and $q(\boldsymbol{x}) = \sum_c q(\boldsymbol{x}|c)$. Let $p_{\boldsymbol{\theta},\boldsymbol{e}}(\boldsymbol{x})$ be a model parameterized by $\boldsymbol{\theta} \in \Theta$ and $\boldsymbol{e} \in E$, where $\boldsymbol{\theta}$ denotes the model parameters and $\boldsymbol{e}$ is the embedding for a condition. In the context of style transfer, let $c_1$, $c_2$, and $c_3$ represent the style reference, the content text in style reference, and the target text prompt, respectively. Models are tasked with generating a plausible target image by combining the content of $c_3$ with the style of $c_1$, while ensuring that the unwanted content $c_2$ does not transfer into the generated result. The corresponding embeddings of $c_1$, $c_2$, and $c_3$ are denoted as $\boldsymbol{e}_1$, $\boldsymbol{e}_2$, and $\boldsymbol{e}_3$. Each of these embeddings is a $d$-dimensional feature. Here, $\boldsymbol{e}_1$ is an image feature that is embedded*

---

[1]We provide the generation results of InstantStyle across various coefficient values in Figure 10 of Appendix A.5, which showcasing that it heavily relies on test-time coefficient-tuning for style strength, requiring users to engage in a labour-intensive process to achieve a balanced synthesis between target content and style.

Figure 3: (a) The proposed content-related elements identification method: we cluster the element-wise product between image and text features and directly discard elements in the high-means cluster; (b) Illustration of tuning-based models, which we detail in Section 3.2. Text-Adapter and Image-Adapter learn the content feature from the text content feature and image feature, respectively. Only the newly added feature adapter modules (denoted as "Linear+LN") are trained while the pre-trained diffusion model is frozen.

by IP-Adapter's image encoder; $e_2$ and $e_3$ are text features embedded by the CLIP model. We denote the element-wise product feature between $e_1$ and $e_2$ as $e_p$, i.e., $e_p^i = e_1^i \cdot e_2^i$.

To mitigate the image-text misalignment issue, we conduct feature manipulation within the latent space of image features. We propose to eliminate content from the style reference by discarding the specific elements corresponding to that content. To achieve this, we drop out these elements by setting their values to zero. As illustrated in Figure 3 (a), the process unfolds as follows:

1) We first compute the element-wise products between the image feature $e_1$ and the corresponding content text feature $e_2$, denoting the results as $e_p^i (i = 1, \cdots, d)$ where $e_p^i = e_1^i \cdot e_2^i$. 2) Next, we cluster these elements $e_p^i (i = 1, \cdots, d)$ into $K$ classes. 3) We then generate a masking vector $m$ based on the clustering results. For the element $e_p^i$ in the highest-means cluster, we set the corresponding element $m^i$ to 0. 4) Finally, we apply the mask vector $m$ to the image feature $e_1$ by computing $e_1' = e_1 \odot m$. This masked image feature $e_1'$, along with the text feature $e_3$, is then incorporated into the cross-attention module of IP-Adapter.

The complete algorithm can be found in Algorithm 1 in Appendix A.3. With Algorithm 1, we explicitly remove the image feature elements that contribute to the similarity between the image feature and content text feature. This means we identify and remove those elements that are most correlated with the content text feature in the feature space, thereby effectively reducing content leakages. Additionally, we provide a detailed explanation of why we utilize clustering on $e_1^i \cdot e_2^i$ to capture content-related elements, rather than relying on the absolute difference between $e_1^i$ and $e_2^i$ or other metrics. Through theoretical exploration, we discovered that masking strategy with on $e_1^i \cdot e_2^i$ leads to the highest energy score for content text feature $e_2$, thus effectively decoupling content from style. We prove this advantage in Proposition 1). Before the proposition, we first give a brief introduction to Energy Diffusion Guidance as follows:

**Energy Diffusion Guidance** Given the noisy image $\mathbf{x}_t$ and the domain of the given conditions $c$, Yu et al. (Yu et al., 2023) proposed a energy diffusion guidance to model the gradient $\nabla_{\mathbf{x}_t} \log p(c|\mathbf{x}_t)$ by resorting to the energy function: $p(c|\mathbf{x}_t) = \frac{\exp(-\lambda \mathcal{E}(c, \mathbf{x}_t))}{Z}$, where $\lambda$ denotes the positive temperature coefficient and $Z > 0$ denotes the normalizing constant, computed as $Z = \int_{c \in \mathcal{C}} \exp\{-\lambda \mathcal{E}(c, \mathbf{x}_t)\}$, $\mathcal{E}(c, \mathbf{x}_t)$ is an energy function that measures the compatibility between the condition $c$ and the noisy image $\mathbf{x_t}$. The value of the energy function will be smaller when $c$ is more compatible with $\mathbf{x_t}$. Therefore, the gradient $\nabla_{\mathbf{x}_t} \log p(c|\mathbf{x}_t)$ can be implemented with the following: $\nabla_{\mathbf{x}_t} \log p(c|\mathbf{x}_t) \propto -\nabla_{\mathbf{x}_t} \mathcal{E}(c, \mathbf{x}_t)$, which is referenced to the *energy guidance*. Following (Yu et al., 2023), we use a time-independent distance measuring functions $\mathcal{D}_{\boldsymbol{\theta}}(c, \mathbf{x}_0)$ to approximate the energy function $\mathcal{E}(c, \mathbf{x}_0)$:

$$\mathcal{E}(c, \mathbf{x}_t) \approx \mathbb{E}_{p(\mathbf{x}_0|\mathbf{x}_t)} \mathcal{D}_{\boldsymbol{\theta}}(c, \mathbf{x}_0) \tag{3}$$

where $\boldsymbol{\theta}$ defines the model parameters of the diffusion model. $\mathcal{D}_{\boldsymbol{\theta}}(c, \mathbf{x}_0)$ computes the cosine similarity between the CLIP embeddings of the given condition $c$ and image $\mathbf{x}_0$.

**Proposition 1** *[The superiority of the proposed masked element selection method]* *We denote the masked elements in the image feature as $e_1^{s+1}, \cdots, e_1^d$ and denote the feature composed by these elements as $e_1^m$, i.e., $e_1^m := [e_1^{s+1}, \cdots, e_1^d]$. Incorporating the masking strategy, i.e., $e_1^m = \varnothing$ leads to $\nabla_{\mathbf{x}_t} \log p(e_1^m|\mathbf{x}_t, e_3) = 0$. According to $\nabla_{\mathbf{x}_t} \log p(c|\mathbf{x}_t) \propto -\nabla_{\mathbf{x}_t} \mathcal{E}(c, \mathbf{x}_t)$, we have approximated*

*the local maximization of $\mathcal{D}_{\boldsymbol{\theta}}(\boldsymbol{e}_1^m, \mathbf{x}_{0|t})$ since $\mathcal{E}(\boldsymbol{e}_1^m, \mathbf{x}_t) \approx \mathcal{D}_{\boldsymbol{\theta}}(\boldsymbol{e}_1^m, \mathbf{x}_{0|t})$. The proposed masking strategy enforces the selected component $\boldsymbol{e}_1^m$ to have the closest distance with the content text feature $\boldsymbol{e}_2$. Therefore, according to the relation between energy and distance as defined in Equation 3, the masking strategy with clustering on $\boldsymbol{e}_1^i \cdot \boldsymbol{e}_2^i$ can lead to the highest energy score for the content text feature $\boldsymbol{e}_2$ when compared to other masking methods.*

This proposition indicates that our proposed masking-based method not only mitigates the image-text misalignment issue by manipulating embeddings within the image feature space, but also achieves the highest energy score for content text feature compared to other masked element selection methods. This effectively reduces the likelihood of content text features, leading to superior performance in content removal.

## 3.2 THEORETICAL EVIDENCE OF FEWER CONDITIONS IN ENHANCING STYLE TRANSFER

In this section, we present the theoretical evidence supporting our method's superiority over InstantStyle and IP-Adapter. Theorem 1 illustrates the advantages of our masking-based approach, suggesting that fewer conditions can achieve a smaller divergence between the generated image distribution and the real image distribution. To delve deeper, we also investigate whether a tuning-based model guided by fewer conditions can yield improved results. Our findings indicate that training models with fewer conditions can also enhance style transfer, as illustrated in Theorem 2.

**Notations 2** *We use $p_{\boldsymbol{\theta}, \boldsymbol{e}}(\boldsymbol{x})$ to approximate the conditional data distribution $q(\boldsymbol{x}|c)$. Let $p_{\boldsymbol{\theta}, \boldsymbol{\phi}}(\boldsymbol{x}|c) = p_{\boldsymbol{\theta}, \boldsymbol{e}}(\boldsymbol{x})|_{\boldsymbol{e}=\boldsymbol{\phi}(c)}$, where $\boldsymbol{e}$ denotes the embedding of the given condition $c$. We denote a certain statistics divergence as $\mathcal{D}$ (or more loosely a divergence upper bound).*

**Theorem 1** *[Why the masking strategy is better] Suppose the divergence $\mathcal{D}$ is convex, and the elements in the image feature are independent of each other. We denote $\boldsymbol{e}_1 := \boldsymbol{e}_1^{i,\cdots,d}$, $\boldsymbol{e}_2 := \boldsymbol{e}_2^{i,\cdots,d}$, and $\boldsymbol{e}_3 := \boldsymbol{e}_3^{i,\cdots,d}$. Thus, the divergence between the generated and ground-truth image distribution of the InstantStyle model is:*

$$D_1 = \mathbb{E}_{q(\boldsymbol{e}_1^{i,\cdots,d}, \boldsymbol{e}_2^{i,\cdots,d}, \boldsymbol{e}_3^{i,\cdots,d})} \mathcal{D}(q(\boldsymbol{x}|\boldsymbol{e}_1, \boldsymbol{e}_2, \boldsymbol{e}_3) \| p_{\boldsymbol{\theta}}(\boldsymbol{x}|\boldsymbol{e}_1^{i,\cdots,d}, \boldsymbol{e}_2^{i,\cdots,d}, \boldsymbol{e}_3^{i,\cdots,d}))$$

*We denote the masked element in the image feature as $\boldsymbol{e}_1^{s+1}, \cdots, \boldsymbol{e}_1^d$. Thus, the divergence result of the proposed masking strategy is:*

$$D_2 = \mathbb{E}_{q(\boldsymbol{e}_1^{i,\cdots,d}, \boldsymbol{e}_2^{i,\cdots,d}, \boldsymbol{e}_3^{i,\cdots,d})} \mathcal{D}(q(\boldsymbol{x}|\boldsymbol{e}_1, \boldsymbol{e}_2, \boldsymbol{e}_3) \| p_{\boldsymbol{\theta}}(\boldsymbol{x}|\boldsymbol{e}_1^{i,\cdots,s}, \boldsymbol{e}_2^{i,\cdots,d}, \boldsymbol{e}_3^{i,\cdots,d}))$$

*With the assumption:*

$$\mathbb{E}_{q(\boldsymbol{e}_1^{i,\cdots,d})} \mathcal{D}(q(\boldsymbol{x}|\boldsymbol{e}_1^{i,\cdots,d}) \| p_{\boldsymbol{\theta}}(\boldsymbol{x}|\boldsymbol{e}_1^{i,\cdots,s})) \leq \mathbb{E}_{q(\boldsymbol{e}_1^{i,\cdots,s})} \mathcal{D}(q(\boldsymbol{x}|\boldsymbol{e}_1^{i,\cdots,s}) \| p_{\boldsymbol{\theta}}(\boldsymbol{x}|\boldsymbol{e}_1^{i,\cdots,s}))$$

*and by Jensen's inequality, we have $D_2 \leq D_1$.*

This theorem indicates that: compared to InstantStyle, masking certain elements (i.e., $\boldsymbol{e}_1^{s+1}, \cdots, \boldsymbol{e}_1^d$) of the image feature achieves a smaller divergence between the generated and ground-truth image distribution. Further, we also investigate whether a tuning-based model conditioned on appropriately fewer conditions can yield improved results. We begin by formalizing the learning paradigms of tuning-based models, as illustrated in Figure 3 (b), as follows:

**Learning Paradigms** *Text-Adapter:* Given the style-reference image $c_1$, the content in style reference $c_2$, and the target text prompt $c_3$, models are tasked with generating images that reflect the text prompt $c_3$ while incorporating the style of $c_1$ and avoiding the presence of content $c_2$. Without loss of generality, we keep the parameters in condition encoders frozen and perform adapter tuning for content text feature $\boldsymbol{e}_2$ with adapter $\phi$. The style reference's image feature $e_1$ is subtracted by the content feature $\phi(e_2)$ to avoid the presence of content $c_2$. The text adapter $\phi$ can be optimized by:

$$\min_{\boldsymbol{\phi}} \mathbb{E}_{q(c_1, c_2, c_3)} \mathcal{D}(q(\boldsymbol{x}|c_1, c_2, c_3) \| p_{\boldsymbol{\theta}, \boldsymbol{\phi}}(\boldsymbol{x}|c_1, c_2, c_3))$$

*Image-Adapter:* Instead of tuning the content text feature, we directly extract the content feature from the style-reference image. We denote the extraction function (image adapter) as $\psi$. Then we

have the final image feature incorporated into the cross-attention module represented as $e_1 - \psi(e_1)$. Thus, the optimization objective is:

$$\min_{\psi} \mathbb{E}_{q(c_1,c_2,c_3)} \mathcal{D}(q(x|c_1, c_2, c_3) \| p_{\theta,\psi}(x|c_1, c_3))$$

In practice, the feature adapters (denoted as "Liner+LN") in Figure 3 (b) are trained through image reconstruction using mean squared error (MSE) loss to predict reconstruction errors. The specific algorithm is provided in Algorithm 2 in Appendix A.3. We instantiate the two optimization objectives, as shown in the 9th line of Algorithm 2, for the Text-Adapter and Image-Adapter, respectively.

**Assumption 1** *Suppose the condition $c_1 \in \mathcal{C}_1$ and $c_3 \in \mathcal{C}_3$ are independent of each other, and the condition $c_2$ is dependent on the style reference $c_1$. Given the condition $c_1 = x$, the content information $c_2 = y$ is uniquely determined.*

Under Assumption 1, we reveal that a tuning-based model conditioned on appropriately fewer conditions can yield improved style transfer results:

**Theorem 2** *[The superiority of tuning-based model conditioned on fewer conditions] Suppose the divergence $\mathcal{D}$ is convex, and the function space $\Phi$ and $\Psi$ ($\phi \in \Phi$ and $\psi \in \Psi$) includes all measurable functions. Under Assumption 1 and by Jensen's inequality, we have:*

$$\min_{\psi} \mathbb{E}_{q(c_1,c_2,c_3)} \mathcal{D}(q(x|c_1, c_2, c_3) \| p_{\theta,\psi}(x|c_1, c_3))$$
$$\leq \min_{\phi} \mathbb{E}_{q(c_1,c_2,c_3)} \mathcal{D}(q(x|c_1, c_2, c_3) \| p_{\theta,\phi}(x|c_1, c_2, c_3)) \tag{4}$$

This theorem indicates that learning content features based on the text feature results in a larger divergence between the generated and real image distribution, compared to learning content features directly within the feature space of image features. Overall, these theoretical results demonstrate that appropriately fewer conditions boost better text-to-image diffusion models in style transfer.

## 4 EXPERIMENTS

In this section, we first demonstrate the proposed theoretical results. Previous evaluation datasets do not contain explicitly defined references' contents, thus making it inaccurate in evaluating content leakages. Instead, we consider an evaluation dataset comprising various defined reference contents and styles for comprehensively assessing models' capability in addressing content leakages. In this paper, we construct the evaluation dataset consisting of 10 content objects and 21 image styles. Extensive experimental results demonstrate that both tuning-free and tuning-based models, conditioned on appropriately selected fewer conditions, achieve higher text fidelity and style similarity, which aligns well with Theorem 1-2. Next, we report our method's performance across various styles and compare it with existing approaches using the StyleBench (Gao et al., 2024) benchmark. Experimental results demonstrate the proposed method's effectiveness in avoiding content leakages.

### 4.1 QUANTITATIVE ANALYSIS OF OUR METHOD IN ADDRESSING CONTENT LEAKAGES

**Evaluation Dataset** We construct the evaluation dataset using the 10 classes of CIFAR-10 (Krizhevsky et al., 2009). Leveraging the code of MACE (Lu et al., 2024), we generate 21 distinct styles for each class, each containing 8 variations. The dataset is divided into two subsets based on image style for training and testing. Using these generated images as references, we train tuning-based models (i.e., Image-Adapter and Text-Adapter) through image reconstruction. During inference, we utilize the test dataset as image references to conduct text-driven style transfer for 5 text prompts. Additional details about the datasets are provided in Table 4 of Appendix A.4.

**Model Configuration** For the tuning-based models, we update adapter weights for 2500 steps using Adam optimizer (Kingma, 2014) with a learning rate of 0.00001. We adopt the same adapter layer structure for Image-Adapter and Text-Adapter, which consists of a linear layer and a batch normalization. Subsequently, leveraging the test data as image references, the trained model is utilized to generate stylized images for 5 text prompts. For all experiments, we adopt Stable Diffusion V1.5 as our base text-to-image model, and we set the clustering classes to 2 for our masking-based method.

Table 1: Quantitative comparison with advanced text-driven style transfer methods. We mark the **Best results** and underscore the second best results.

| Method | StyleCrafter | StyleAligned | StyleDrop | DEADiff | InstantStyle | StyleShot | Ours |
|---|---|---|---|---|---|---|---|
| style score ↑ | 0.245 | 0.244 | 0.240 | 0.230 | **0.290** | 0.267 | 0.273 |
| fidelity score ↑ | 0.858 | 0.662 | 0.889 | 0.916 | 0.820 | 0.956 | **0.972** |
| leakage score ↓ | 0.589 | 0.720 | 0.600 | 0.523 | 0.596 | 0.543 | **0.478** |
| Human Preference ↑ | 1.7% | 4.7% | 19.6% | 4.7% | 8.3 % | 24.3% | **36.7%** |

**Evaluation Metrics** *Human Preference*: Following (Liu et al., 2023; Qi et al., 2024), we conduct user preference studies to evaluate models' style transfer ability. Compared to other methods, our method achieves the highest human preferences by a large margin, demonstrating robust stylization across various styles and responsiveness to text prompts. *CLIP-Based Scores*: We also assess the quality of generated images using CLIP (Radford et al., 2021) with ViT-H/14 as the image encoder. We perform binary classification using the CLIP model Radford et al. (2021) on the generated images to distinguish between the reference's content text and the text prompt. The computed classification accuracy is referred to as the **fidelity score** ↑. We also calculate the similarity between the generated images and the reference's content text, calibrated by the similarity between the reference images and the content text, termed the **leakage score** ↓. Finally, we assess the similarity between the generated images and the style reference, adjusted by subtracting the leakage score, which we refer to as the **style score** ↑. The fidelity score measures the fidelity to the text instructions, the style score assesses style similarity with the style reference, and the leakage score indicates content leakages from the style reference, where a lower score is preferable. More details of these scores are provided in Appendix A.4.

**Experiment Results** We provide visual comparisons between the proposed masking-based method and state-of-the-art methods in Figure 4, demonstrating that our approach can mitigate content leakages without introducing style degradations. These results align well with the theoretical findings of Theorem 1, showcasing that fewer conditions more effectively address both content leakages and style degradations.

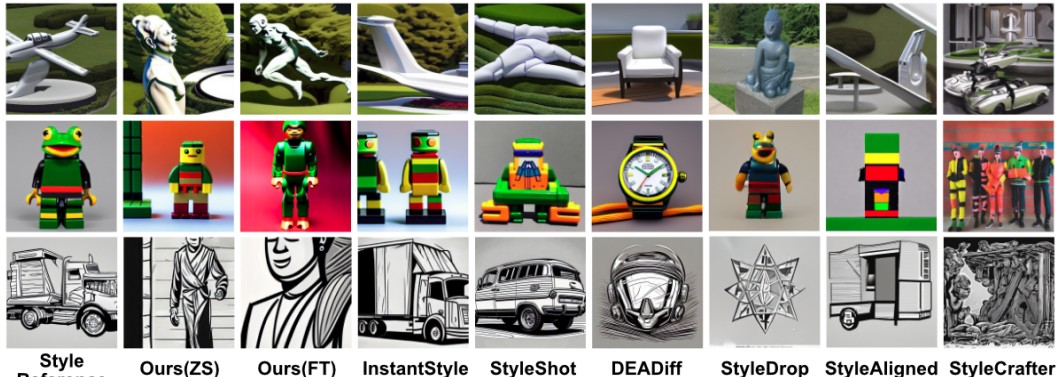

Figure 4: In the figure, the text prompt is "A human". Leveraging appropriately fewer conditions, Ours(ZS) and Ours(FT) denote the proposed masking-based method and the tuning-based Image-Adapter method, respectively. **Our methods successfully transfer the references' styles without content leakages.** More results can be found in Figure 9 and Figure 12-16 in Appendix A.6.

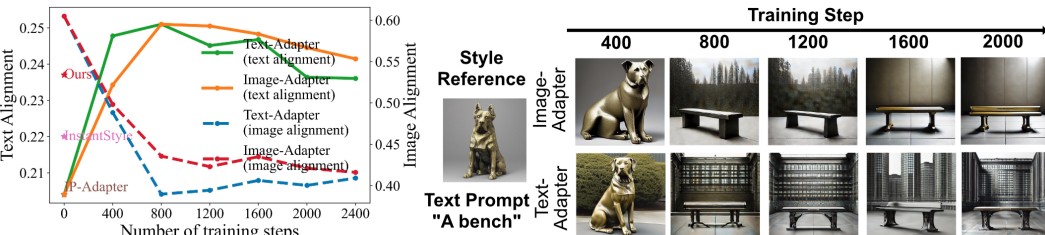

(a) Image and text alignment results during training     (b) Visualization results of Image-Adapter and Text-Adapter

Figure 5: Comparison between the Image-Adapter and Text-Adapter model. (a) Following (Gao et al., 2024), we report the image and text alignment scores alongside training steps. We also present the tuning-free models' (i.e., IP-Adapter, InstantStyle, and our masking-based method) fidelity scores in the figure. (b) Visual comparisons between Image-Adapter and Text-Adapter.

Table 2: Quantitative comparison with state-of-the-art text-driven style transfer methods. The highest human preference scores demonstrate the effectiveness of our method.

| Method | StyleCrafter | DEADiff | StyleDrop | InST | StyleAligned | StyleShot | InstantStyle | Ours |
|---|---|---|---|---|---|---|---|---|
| text alignment ↑ | 0.202 | 0.232 | 0.220 | 0.204 | 0.213 | 0.219 | 0.275 | 0.265 |
| image alignment ↑ | 0.706 | 0.597 | 0.621 | 0.623 | 0.680 | 0.640 | 0.575 | 0.657 |
| Human Preference ↑ | 4.2% | 10.1% | 2.6% | 5.7% | 6.3% | 21.1% | 7.9% | **42.1%** |

For the tuning-based models, we present their image and text alignment scores and generation results along with the training steps in Figure 5. The observations are as follows: 1) Guided by appropriately-selected fewer conditions, the Image-Adapter model outperforms the Text-Adapter in both style scores and fidelity scores, indicating a smaller distribution divergence between the generated images and real images, consistent with the theoretical results of Theorem 2. 2) As shown in Figure 5 (b), while the Text-Adapter reduces content leakages, it leads to significant style degradation as the training steps increase. In contrast, by leveraging fewer conditions, the Image-Adapter successfully avoids the image and text modal misalignment, with no content leakages while achieving style enhancement.

Overall, we provide quantitative comparisons with recent methods, such as StyleCrafter (Liu et al., 2023), StyleAligned (Hertz et al., 2024), StyleDrop (Sohn et al., 2024), DEADiff (Qi et al., 2024), StyleShot (Gao et al., 2024) and so on. We present the CLIP-based score results in Table 1. Additional visual comparisons can be found in Figure 9 and Figure 12-16 of Appendix A.6. The overall best performances in CLIP-based scores and human preferences further demonstrate the effectiveness of our method in balancing content leakage mitigation with style enhancement.

## 4.2 COMPARISON WITH STATE-OF-THE-ART METHODS ON STYLEBENCH

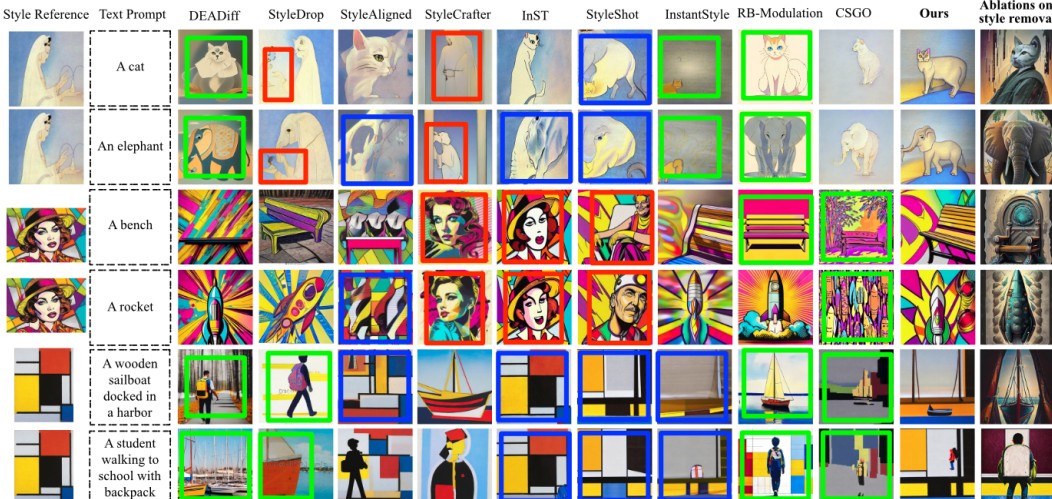

Figure 6: Visual comparison between recent state-of-the-art methods, including RB-Modulation (Rout et al., 2024), CSGO (Xing et al., 2024), for addressing content leakage, and ours in text-driven style transfer. We mark the results with significant content leakages, style degradation, and loss of text fidelity with red, green, and blue boxes, respectively. The proposed masking-based method does not require content knowledge of the image reference; instead, we leverage the CLIP text feature of "person, animal, plant, or object in the foreground" to identify the elements that need to be masked. More comparison examples are in Figure 11 of Appendix A.6.

**Experiment Details** To comprehensively evaluate the effectiveness of the proposed masking-based method, we conduct evaluations on the recent style transfer benchmark StyleBench (Gao et al., 2024), which covers 73 distinct styles, ranging from paintings, flat illustrations to sculptures with varying materials. For InstantStyle and our method, we employ the feature subtraction and masking strategy, respectively, on the extracted image features by StyleShot.

**Experiment Results** Following StyleShot (Gao et al., 2024), we report the quantitative comparison on text and image alignment with state-of-the-art text-driven style transfer methods in Table 2. Figure 6 displays our results and baselines of four distinct style images, each corresponding to the same

pair of text prompts. As shown in Figure 6, we observe that InstantStyle (Wang et al., 2024a) and the most recent method StyleShot (Gao et al., 2024) retain the image style but may fail to generate the target semantic information. In contrast, our method can improve text fidelity for text prompts without sacrificing style enhancement, avoiding content leakages and achieving style enhancement. As shown in the last column of Figure 6, we also present ablation study results in the last column, where we retain the identified elements to be discarded while masking the other features. Consequently, there is almost no style information identical to the reference image, further confirming that our method can efficiently and accurately decouple content from style.

### 4.3 ABLATION STUDIES

**Effectiveness in Image-Driven Style Transfer**
The proposed method also excels at transferring style onto target content images. We compare our method with the recent SOTA method StyleShot (Gao et al., 2024) to showcase the superiority of our method in efficiently mitigating content leakages. As shown in Figure 7, StyleShot usually generates unsatisfied results when the style reference image consists of a human face. In contrast, according to the clustering results on the element-wise product feature $e_p$, by only masking 1-5 elements, our method can successfully mitigate content leakages and style degradation of StyleShot.

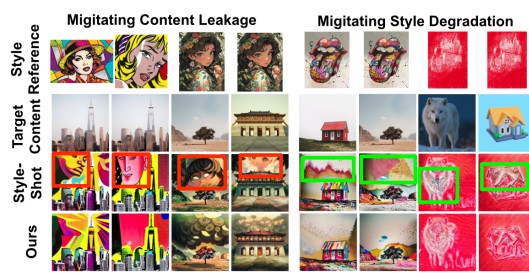

Figure 7: Visual comparison between StyleShot and ours in image-driven style transfer. Results with content leakages and style degradation are highlighted with red and green boxes, respectively. More results are in Figure 19.

Moreover, we also compare our method with InstantStyle's block-specific injection technique based on different diffusion models and style encoders. Due to space limitations, we provide additional results in Appendix A.7.

**Ablation Studies on Clustering Number** In Table 3, we conduct ablation studies clustering number $K$ in text-driven style transfer using the StyleBench dataset. Due to space limitations, we provide additional visualization results in Appendix A.8. The results show that a smaller $K$, such as $K = 2$, can lead to higher text alignment scores, as more content-related elements in the style reference are masked. This is particularly evident in styles such as 3D models, Anime, and Baroque art, which contain more human-related images, resulting in more effective content leakage avoidance.

## 5 CONCLUSION

Table 3: Ablation study results on clustering number $K$.

|  | $K$ | 3D Model | Anime | Baroque |
|---|---|---|---|---|
| image alignment | 2 | 0.474 | 0.372 | 0.384 |
| | 3 | 0.478 | 0.381 | 0.393 |
| | 4 | 0.485 | 0.390 | 0.404 |
| | 5 | 0.487 | 0.380 | 0.411 |
| text alignment | 2 | 0.213 | 0.234 | 0.257 |
| | 3 | 0.206 | 0.232 | 0.253 |
| | 4 | 0.189 | 0.231 | 0.253 |
| | 5 | 0.188 | 0.229 | 0.252 |

In this paper, we propose a masking-based method, that efficiently decouples content from style without requiring tuning any model parameters. By masking (zeroing out) certain elements in the image feature corresponding to that content, we effectively eliminate content leakages from style references across various evaluation datasets. More importantly, we have theoretically proved that our model, under the guidance of appropriately selected fewer conditions, achieves a smaller divergence between the generated image distribution and the real image distribution, outperforming those conditioned on larger, yet less coherent conditions. Extensive experiments across various styles and targets have demonstrated the effectiveness of our proposed method.

## 6 ETHICS STATEMENT

This work aims to make a positive impact on the field of AI-driven image generation. Our method builds upon existing open-source diffusion models, which already include some built-in safeguards, such as the NSFW ("not safe for work") Filter. We believe that our contributions in generating images with diverse styles, while preventing content leakages, can further positively help avoid stereotypes and discriminations. Further details regarding our ethics statement are provided in Appendix A.11

ACKNOWLEDGEMENTS

This work is supported by National Natural Science Foundation of China under Grant (No.62106139).

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

## A APPENDIX

### A.1 RELATED WORKS

**Stylized Image Generation** Stylized image generation, commonly referred to as image style transfer, involves transferring the stylistic or aesthetic attributes from a reference image to a target image. Thanks to the significant advancements in diffusion models (Ho et al., 2020; Podell et al., 2023; Song et al., 2022; Li et al., 2024; Rombach et al., 2022; Ho & Salimans, 2022; Ramesh et al., 2022; Saharia et al., 2022; Nichol et al., 2021), numerous methods (Sun et al., 2023; Xu et al., 2024; Lu et al., 2023; Lin et al., 2024) have been developed to ensure style consistency across images generated. Among inversion-based approaches (Zhang et al., 2023c; Gal et al., 2022; Hertz et al., 2024) project style images into a learnable embedding in the text token space to guide style-specific generation. Unfortunately, these methods can lead to information loss due to the mapping from visual to text modalities. Cross-attention manipulation (Le & Carlsson, 2022; Hertz et al., 2024; Chung et al., 2024; Hertz et al., 2024; Wang et al., 2023b) is another method for achieving style transfer, involving the manipulation of features within self-attention layers. In contrast, IP-Adapter (Ye et al., 2023) and Style-Adapter (Wang et al., 2023b) introduce a distinct cross-attention mechanism that de-couples the attention layers for text and image features, allowing for a coarse control over the style transfer process. Although these methods have achieved significant advancements, they often struggle with content leakages from style-reference images.

**Methods Addressing Content Leakages** Some approaches (Zhang et al., 2018a;b; Qi et al., 2024) aim to tackle the content leakages issue by constructing paired datasets where images share the same subject matter but exhibit distinct styles, facilitating the extraction of disentangled style and content representations. DEADiff (Qi et al., 2024) stands out by extracting disentangled representations of content and style using a paired dataset, facilitated by the Q-Former (Li et al., 2023) technique. Other works (Sohn et al., 2024; Liu et al., 2023) optimize some or all model parameters using extensive style images, embedding their visual style into the model's output domain. However, the inherently underdetermined nature of style makes the creation of large-scale paired datasets or style datasets both resource-intensive and limited in the diversity of styles it can capture. To address this issue, InstantStyle (Wang et al., 2024a), a recent innovation, employs block-specific injection and feature subtraction techniques to implicitly achieve decoupling of content and style, offering a nuanced approach to style transfer. In the context of image-driven style transfer, InstantStyle-Plus (Wang et al., 2024b) further proposed several techniques to prioritize the integrity of the content image while seamlessly integrating the target style. Although the InstantStyle approach achieved significant advancements, feature manipulation across different modalities inevitably introduces the image-text misalignment issue (Kim et al., 2023; Gordon et al., 2023), which hinders accurate disentanglement of content and style. StyleDiffusion (Wang et al., 2023a) introduced a CLIP-based style disentanglement loss coordinated with a style reconstruction to decouple content from style in the CLIP image space. However, this framework required a training process to disentangle style from each style image, achieving this by providing approximately 50 content images for training. DiffuseIT (Kwon & Ye, 2022) introduced a novel diffusion-based unsupervised image translation method for decoupling content from style, but it also requires complex loss regularization. More recent and stronger models, such as RB-Modulation (Rout et al., 2024), have been proposed to alleviate the content leakage problem. RB-Modulation uses attention-based feature aggregation and different descriptors to decouple content and style. It is training-free and is reported to outperform InstantStyle. CSGO (Xing et al., 2024) is another recent approach that employs a separately trained style projection layer to mitigate content leakage. Additionally, Zhao et al (Zhao et al., 2024a) proposed a method to identify and address the issue of conditional content leakage in image-to-video (I2V) generation. Several studies (Motamed et al., 2023; Huang et al., 2024; Le & Carlsson, 2022) focus on concept disentanglement, but they are not specifically aimed at style transfer.

**Masking Mechanism in Synthesizing High-Quality Images** Although several studies (Couairon et al., 2022; Gao et al., 2023; Hansen-Estruch et al., 2024; Pan et al., 2023; Lei et al., 2023) have explored the effectiveness of masking mechanisms, our method differs from these approaches in several key aspects: 1) No coupled denoising processes: Our method avoids the need for two denoising processes, thus saving computational resources. For instance, the DIFFEDIT method (Couairon et al., 2022) requires two denoising processes—one conditioned on the query text and the other conditioned on a reference text. By contrasting the predictions of the two diffusion models, DIFFEDIT generates a mask that locates the regions needing editing to match the query text. 2) Masking in the

latent space: Unlike DIFFEDIT (Couairon et al., 2022), which operates on the pixel level to generate a mask highlighting the regions of the input image that need editing, our method performs masking in the latent space, bypassing pixel-level operations and patch-level manipulations. 3) Focus on content leakage in style transfer: While the MDT method (Gao et al., 2023) introduces a latent masking scheme to enhance the DPMs' ability to learn contextual relations among object semantics in an image, it focuses on predicting randomly masked tokens from unmasked ones. In contrast, our method targets content leakage in style transfer. We mask feature elements that are related to unwanted content from the style reference, guided by clustering results on the element-wise product.

## A.2 PROOF

**Proof of Proposition 1** We denote the masked elements in the image feature as $e_1^{s+1}, \cdots, e_1^d$ and denote the feature composed by these elements as $e_1^m$, i.e., $e_1^m := [e_1^{s+1}, \cdots, e_1^d]$. Following (Yu et al., 2023), we use a time-independent distance measuring functions $\mathcal{D}_{\boldsymbol{\theta}}(c, \mathbf{x}_0)$ to approximate the energy function $\mathcal{E}(c, \mathbf{x}_0)$:

$$\mathcal{E}(c, \mathbf{x}_t) \approx \mathbb{E}_{p(\mathbf{x}_0|\mathbf{x}_t)} \mathcal{D}_{\boldsymbol{\theta}}(c, \mathbf{x}_0) \tag{5}$$

where $\boldsymbol{\theta}$ defines the model parameters of the diffusion model. $\mathcal{D}_{\boldsymbol{\theta}}(c, \mathbf{x}_0)$ computes the cosine similarity between the CLIP embeddings of the given condition $c$ and image $\mathbf{x}_0$.

Based on the classifier-free guidance (Ho & Salimans, 2022), incorporating the masking strategy, i.e., $e_1^m = \varnothing$ leads to $\nabla_{\mathbf{x}_t} \log p(e_1^m | \mathbf{x}_t, e_3) = 0$.

Building upon the energy-based assumption in (Yu et al., 2023), i.e., $\nabla_{\mathbf{x}_t} \log p(c|\mathbf{x}_t) \propto -\nabla_{\mathbf{x}_t} \mathcal{E}(c, \mathbf{x}_t)$, we have approximated the local maximization of $\mathcal{E}(e_1^m, \mathbf{x}_t)$. Again since $\mathcal{E}(e_1^m, \mathbf{x}_t) \approx \mathcal{D}_{\boldsymbol{\theta}}(e_1^m, \mathbf{x}_{0|t})$, we have approximated the local maximization of $\mathcal{D}_{\boldsymbol{\theta}}(e_1^m, \mathbf{x}_{0|t})$.

Based on the clustering result on element-wise product feature $e_p(e_p^i = e_1^i \cdot e_2^i)$, we mask (drop out) the high-value elements, denoted as $e_1^m$, in feature $e_p$. With a fixed masking proportion, our proposed strategy differs from other methods—such as those relying on the absolute difference between $e_1^i$ and $e_2^i$—by ensuring that the selected component $[0, \cdots, 0, e_1^{s+1}, \cdots, e_1^d]$ have the highest cosine similarity with the content text feature $e_2$. This approach can lead to $\max \mathcal{D}_{\boldsymbol{\theta}}(e_2, \mathbf{x}_{0|t})$ when compared to other masking methods.

Note that the energy function for $e_2$ satisfies: $\mathcal{E}(e_2, \mathbf{x}_t) = \mathcal{D}_{\boldsymbol{\theta}}(e_2, \mathbf{x}_{0|t})$. Therefore, according to the relation between energy and distance as defined in Equation 3, the masking strategy with clustering on $e_1^i \cdot e_2^i$ can lead to the highest energy score for the content text feature $e_2$.

**Proof of Theorem 1** We denote the element in the image feature and text feature as $e_1^i (i \in \{1, \cdots, d\})$ and $e_2^i (i \in \{1, \cdots, d\})$, respectively. Thus, the divergence between the generated and ground-truth image distribution of the InstantStyle model is:

$$D_1 = \mathbb{E}_{q(e_1^{i,\cdots,d}, e_2^{i,\cdots,d}, e_3^{i,\cdots,d})} \mathcal{D}(q(\boldsymbol{x}|e_1, e_2, e_3) \| p_{\boldsymbol{\theta}}(\boldsymbol{x}|e_1^{i,\cdots,d}, e_2^{i,\cdots,d}, e_3^{i,\cdots,d}))$$

We denote the masked element in the image feature as $e_1^{s+1}, \cdots, e_1^d$, and the divergence result of the proposed masking strategy is:

$$D_2 = \mathbb{E}_{q(e_1^{i,\cdots,d}, e_2^{i,\cdots,d}, e_3^{i,\cdots,d})} \mathcal{D}(q(\boldsymbol{x}|e_1, e_2, e_3) \| p_{\boldsymbol{\theta}}(\boldsymbol{x}|e_1^{i,\cdots,s}, e_2^{i,\cdots,d}, e_3^{i,\cdots,d}))$$

With the assumption:

$$\mathbb{E}_{q(e_1^{i,\cdots,d})} \mathcal{D}(q(\boldsymbol{x}|e_1^{i,\cdots,d}) \| p_{\theta}(\boldsymbol{x}|e_1^{i,\cdots,s})) \leq \mathbb{E}_{q(e_1^{i,\cdots,s})} \mathcal{D}(q(\boldsymbol{x}|e_1^{i,\cdots,s}) \| p_{\theta}(\boldsymbol{x}|e_1^{i,\cdots,s}))$$

and by Jensen's inequality (Ash & Doléans-Dade, 2000), we have

$$\mathbb{E}_{q(e_1^{i,\cdots,d}, e_2^{i,\cdots,d}, e_3^{i,\cdots,d})} \mathcal{D}(q(\boldsymbol{x}|e_1, e_2, e_3) \| p_{\boldsymbol{\theta}}(\boldsymbol{x}|e_1^{i,\cdots,s}, e_2^{i,\cdots,d}, e_3^{i,\cdots,d}))$$

$$\leq \mathbb{E}_{q(e_1^{i,\cdots,s}, e_2^{i,\cdots,d}, e_3^{i,\cdots,d})} \mathcal{D}(\mathbb{E}_{q(e_1^{s+1,\cdots,d})} q(\boldsymbol{x}|e_1, e_2, e_3) \| \mathbb{E}_{q(e_1^{s+1,\cdots,d})} p_{\boldsymbol{\theta}}(\boldsymbol{x}|e_1^{i,\cdots,d}, e_2^{i,\cdots,d}, e_3^{i,\cdots,d}))$$

$$\leq \mathbb{E}_{q(e_1^{s+1,\cdots,d})} \mathbb{E}_{q(e_1^{i,\cdots,s}, e_2^{i,\cdots,d}, e_3^{i,\cdots,d})} \mathcal{D}(q(\boldsymbol{x}|e_1, e_2, e_3) \| p_{\boldsymbol{\theta}}(\boldsymbol{x}|e_1^{i,\cdots,d}, e_2^{i,\cdots,d}, e_3^{i,\cdots,d}))$$

$$= \mathbb{E}_{q(e_1^{i,\cdots,d}, e_2^{i,\cdots,d}, e_3^{i,\cdots,d})} \mathcal{D}(q(\boldsymbol{x}|e_1, e_2, e_3) \| p_{\boldsymbol{\theta}}(\boldsymbol{x}|e_1^{i,\cdots,d}, e_2^{i,\cdots,d}, e_3^{i,\cdots,d}))$$

$$\tag{6}$$

The last line is because of the assumption that the elements of the image feature are independent of each other. Thus we complete the proof of $D_2 \leq D_1$.

**Proof of Theorem 2** Suppose the divergence $\mathcal{D}$ is convex, and the function space $\Phi$ and $\Psi$ ($\phi \in \Phi$ and $\psi \in \Psi$) includes all measurable functions. Under Assumption 1 and by Jensen's inequality, we have:

$$\begin{aligned}
\mathcal{D}(q(\boldsymbol{x}|c_1, c_3) \| p_{\boldsymbol{\theta}, \boldsymbol{\psi}}(\boldsymbol{x}|c_1, c_3)) &= \mathcal{D}(\mathbb{E}_{q(c_2)} q(\boldsymbol{x}|c_1, c_2, c_3) \| \mathbb{E}_{q(c_2)} p_{\boldsymbol{\theta}, \boldsymbol{\psi}}(\boldsymbol{x}|c_1, c_2, c_3)) \\
&\leq \mathbb{E}_{q(c_2)} \mathcal{D}(q(\boldsymbol{x}|c_1, c_2, c_3) \| p_{\boldsymbol{\theta}, \boldsymbol{\psi}}(\boldsymbol{x}|c_1, c_2, c_3))
\end{aligned} \tag{7}$$

Under Assumption 1, the condition $c_1 \in \mathcal{C}_1$ and $c_3 \in \mathcal{C}_3$ are independent of each other, and the condition $c_2$ is dependent on the style reference $c_1$. Given the condition $c_1 = x$, the content information $c_2 = y$ is uniquely determined. Thus, we have $q(c_2)q(c_1, c_3) \geq q(c_1, c_2, c_3)$, since $q(c_1 = x, c_2 = y, c_3 = z) = q(c_1 = x, c_3 = z)$ and $q(c_1 = x, c_2 \neq y, c_3 = z) = 0$.

By the Tower Law (Ash & Doléans-Dade, 2000) and non-negativity of $\mathcal{D}$, we have

$$\begin{aligned}
&\mathbb{E}_{q(c_1, c_2, c_3)} \mathcal{D}(q(\boldsymbol{x}|c_1, c_2, c_3) \| p_{\boldsymbol{\theta}, \boldsymbol{\psi}}(\boldsymbol{x}|c_1, c_3)) \\
&\leq \mathbb{E}_{q(c_2|c_1, c_3)} \mathbb{E}_{q(c_1, c_3)} \mathcal{D}(q(\boldsymbol{x}|c_1, c_2, c_3) \| p_{\boldsymbol{\theta}, \boldsymbol{\psi}}(\boldsymbol{x}|c_1, c_3)) \\
&= \mathbb{E}_{q(c_1, c_3)} \mathcal{D}(q(\boldsymbol{x}|c_1, c_3) \| p_{\boldsymbol{\theta}, \boldsymbol{\psi}}(\boldsymbol{x}|c_1, c_3))
\end{aligned} \tag{8}$$

It is straightforward to extend this to the minimization case, leading to the following inequality:

$$\min_{\boldsymbol{\psi}} \mathbb{E}_{q(c_1, c_2, c_3)} \mathcal{D}(q(\boldsymbol{x}|c_1, c_2, c_3) \| p_{\boldsymbol{\theta}, \boldsymbol{\psi}}(\boldsymbol{x}|c_1, c_3)) \leq \min_{\boldsymbol{\psi}} \mathbb{E}_{q(c_1, c_3)} \mathcal{D}(q(\boldsymbol{x}|c_1, c_3) \| p_{\boldsymbol{\theta}, \boldsymbol{\psi}}(\boldsymbol{x}|c_1, c_3)) \tag{9}$$

Combining Equation 7 and Equation 9, we have:

$$\begin{aligned}
&\min_{\boldsymbol{\psi}} \mathbb{E}_{q(c_1, c_2, c_3)} \mathcal{D}(q(\boldsymbol{x}|c_1, c_2, c_3) \| p_{\boldsymbol{\theta}, \boldsymbol{\psi}}(\boldsymbol{x}|c_1, c_3)) \\
&\leq \min_{\boldsymbol{\phi}} \mathbb{E}_{q(c_1, c_2, c_3)} \mathcal{D}(q(\boldsymbol{x}|c_1, c_2, c_3) \| p_{\boldsymbol{\theta}, \boldsymbol{\phi}}(\boldsymbol{x}|c_1, c_2, c_3))
\end{aligned} \tag{10}$$

Thus complete the proof.

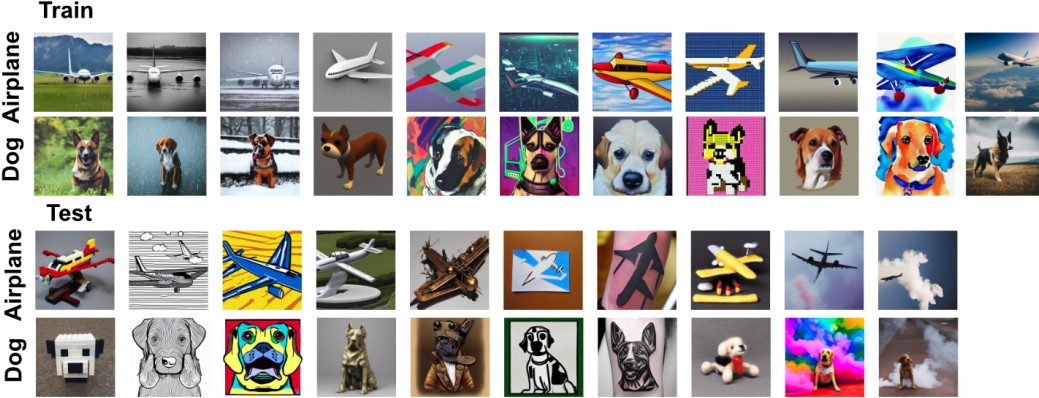

Figure 8: Visual examples of the style-reference images in our constructed dataset. The training sets are only used for learning Image-Adapter and Text-Adapter.

## A.3 ALGORITHM

As shown in the 9th line of Algorithm 2, we highlight the optimization objective for the Text-Adapter and Image-Adapter, respectively. These optimization objectives are used to minimize the prediction error of noise in reconstructing style reference while maximizing the difference between the model conditioned on the style reference's content and that conditioned on the target prompt.

---

**Algorithm 1** Algorithm of the proposed Tuning-Free masking-based method

---

1: **Input:** the style reference $c_1$, default style reference's content text prompt $c_2$, target text prompt $c_3$, VAE-encoder $\mathcal{E}$, pre-defined parameters $\alpha_t$, the repeat times of time travel $T$.
2: **for** $t = 1$ **to** $T$ **do**
3:    $\boldsymbol{\epsilon} \sim \mathcal{N}(\mathbf{0}, \mathbf{I})$, $\mathbf{z}_0 = \mathcal{E}(c_1)$
4:    $\mathbf{z}_t = \alpha_t \mathbf{z_{t-1}} + (1 - \alpha_t)\boldsymbol{\epsilon}$    ▷ Add noise to the latent feature
5: **end for**
6: **for** $t = T$ **to** $1$ **do**
7:    Calculate noise prediction (InstantStyle performs (a) and ours performs (b)):
     (a) $\boldsymbol{\epsilon}_\theta(\mathbf{z}_t, t, \boldsymbol{e}_1 - \boldsymbol{e}_2, \boldsymbol{e}_3)$    (InstantStyle)
     (b) Perform clustering (such as K-means clustering (Jin & Han, 2010)) on element product $\boldsymbol{e}_1^i \cdot \boldsymbol{e}_2^i$; Generate masking vector $\mathbf{m}$, where we set the element value to 1 for those elements in the highest-means cluster; Calculate $\boldsymbol{\epsilon}_\theta(\mathbf{z}_t, t, \boldsymbol{e}_1 \odot \mathbf{m}, \boldsymbol{e}_3)$    (Ours)
8:    Denoise diffusion model using predicted noise;
9: **end for**
10: **Output:** decode the reversed latent code $\mathbf{z}_0$ to image space and output the generated image

---

**Algorithm 2** Algorithm of the Tuning-Based model: Image-Adapter and Text-Adapter

---

1: **Input:** the data distribution of style reference $c_1$, style reference's content text prompt $c_2$ and target text prompt $c_3$, text adapter $\boldsymbol{\phi}_{\theta_1}(\cdot)$, image adapter $\boldsymbol{\psi}_{\theta_2}(\cdot)$, VAE-encoder $\mathcal{E}$, pre-defined parameters $\alpha_t$, hyper-parameter $\lambda$, maximum training step $M$ and the repeat times of time travel $T$.
2: **for** $m = 1$ **to** $M$ **do**
3:    $(c_1, c_2, c_3) \sim p(c_1, c_2, c_3)$    ▷ Sample conditions from data distribution
4:    **for** $t = 1$ **to** $T$ **do**
5:      $\boldsymbol{\epsilon} \sim \mathcal{N}(\mathbf{0}, \mathbf{I})$, $\mathbf{z}_0 = \mathcal{E}(c_1)$
6:      $\mathbf{z}_t = \alpha_t \mathbf{z_{t-1}} + (1 - \alpha_t)\boldsymbol{\epsilon}$    ▷ Add noise to the latent feature
7:    **end for**
8:    **for** $t = T$ **to** $1$ **do**
9:      Take the gradient step (Text-Adapter performs (a) and Image-Adapter performs (b)):
      (a) $\nabla_{\theta_1}[\|\boldsymbol{\epsilon}_\theta(\mathbf{z}_t, t, \boldsymbol{e}_1 - \boldsymbol{\phi}_{\theta_1}(\boldsymbol{e}_2), \boldsymbol{e}_3) - \boldsymbol{\epsilon}\|^2 - \lambda\|\boldsymbol{\epsilon}_\theta(\mathbf{z}_t, t, \boldsymbol{e}_1 - \boldsymbol{\phi}_{\theta_1}(\boldsymbol{e}_2), \boldsymbol{e}_3) - \boldsymbol{\epsilon}_\theta(\mathbf{z}_t, t, \boldsymbol{e}_1 - \boldsymbol{\phi}_{\theta_1}(\boldsymbol{e}_2), \boldsymbol{e}_2)\|^2]$    (Text-Adapter)
      (b) $\nabla_{\theta_2}[\|\boldsymbol{\epsilon}_\theta(\mathbf{z}_t, t, \boldsymbol{e}_1 - \boldsymbol{\psi}_{\theta_2}(\boldsymbol{e}_1), \boldsymbol{e}_3) - \boldsymbol{\epsilon}\|^2 - \lambda\|\boldsymbol{\epsilon}_\theta(\mathbf{z}_t, t, \boldsymbol{e}_1 - \boldsymbol{\psi}_{\theta_2}(\boldsymbol{e}_1), \boldsymbol{e}_3) - \boldsymbol{\epsilon}_\theta(\mathbf{z}_t, t, \boldsymbol{e}_1 - \boldsymbol{\psi}_{\theta_2}(\boldsymbol{e}_1), \boldsymbol{e}_2)\|^2]$    (Image-Adapter)
10:      Update the model parameter $\theta_1$ or $\theta_2$
11:    **end for**
12: **end for**
13: **Output:** The Image-Adapter and Text-Adapter model

---

## A.4   Evaluation Datasets and Metrics

### A.4.1   Evaluation Datasets

Previous evaluation datasets do not contain explicitly defined references' contents, thus making it inaccurate in evaluating content leakages. Instead, we consider an evaluation dataset comprising various defined reference contents and styles for comprehensively assessing models' capability in addressing content leakages. In this paper, we construct the evaluation dataset consisting of 10 content objects and 21 image styles. We present detailed information on the constructed dataset in Table 4. Visual examples of the style-reference images are provided in Figure 8.

### A.4.2   Evaluation Metrics

Following previous studies (Sohn et al., 2024; Gao et al., 2024), we report the image alignment and text alignment scores in Table 2 and Figure 5.

**Image alignment and text alignment** ↑: Image alignment refers to the cosine similarity between the CLIP embeddings of the generated images and the style reference images, while text alignment

measures the cosine similarity between the CLIP embeddings of the generated images and the target text prompts.

Our method is specifically designed to address content leakage in style transfer diffusion models. In addition to the image alignment and text alignment scores used in previous studies (Sohn et al., 2024; Gao et al., 2024), we also introduce three quantitative metrics to comprehensively assess the quality of the generated images from the perspectives of style similarity, text fidelity, and content leakages from style references. We denote the CLIP image feature of the generated image and the style-reference image as $e_g$ and $e_1$, respectively. Given the class name of style reference's content object and the target text prompt, we denote the CLIP text feature of reference's content object and text prompt as $e_2$ and $e_3$, respectively.

**Fidelity score** $\uparrow$: We perform binary classification on the generated images to differentiate between the reference's content object and the text prompt, computing the classification accuracy, referred to as the fidelity score $\uparrow$. Specifically, we denote the cosine similarity between the CLIP image feature of the generated image and the CLIP text feature of reference's content object as $\frac{\langle e_2, e_g \rangle}{|e_2| \cdot |e_g|}$. Similarly, we denote the cosine similarity between the CLIP image feature of the generated image and the CLIP text feature of text prompt as $\frac{\langle e_3, e_g \rangle}{|e_3| \cdot |e_g|}$. If $\frac{\langle e_2, e_g \rangle}{|e_2| \cdot |e_g|} < \frac{\langle e_3, e_g \rangle}{|e_3| \cdot |e_g|}$, the generated image is considered correctly classified, meaning it contains the target content rather than the content of the style reference. Therefore, the fidelity score mainly reflects the models' control ability of text prompts.

**Leakage score** $\downarrow$: We calculate the similarity between the generated images and the reference's content text, calibrated by the similarity between the reference images and the content text, termed the leakage score $\downarrow$. Thus, the leakage score is calculated by:

$$\begin{cases} < e_g, e_2 > / < e_1, e_2 > & \text{if } e_g \text{ is accurately classified} \\ 1 & \text{else} \end{cases} \tag{11}$$

The leakage score indicates content leakages from the style reference, where a lower score is preferable.

**Style score** $\uparrow$: Finally, we assess the similarity between the generated images and the style reference, adjusting by subtracting the similarity between the generated images and the reference's content text, which we refer to as the style score $\uparrow$. The style score is calculated as follows:

$$\begin{cases} < e_g, e_1 > - < e_g, e_2 > & \text{if } e_g \text{ is accurately classified} \\ 0 & \text{else} \end{cases} \tag{12}$$

Compared to the image alignment score, the style score more accurately reflects the style similarity between the generated images and the style references.

**Human preference score** $\uparrow$: In addition to objective evaluations, we have also designed a user study to subjectively assess the practical performance of various methods. In Section 4.1, the constructed dataset consists of 10 content objects (from CIFAR-10) and 21 image styles (11 for training and 10 for testing) for each content object, with 8 variations per style. This results in a total of $10 \times 11 \times 8 = 880$ style references. For each style reference, we perform style transfer for 5 target text prompts, with 4 generations per target text prompt, leading to $880 \times 4 = 3520$ generations per text prompt. We randomly sample 50 images from the 3520 generated images for each target text prompt. In total, this gives us $50 \times 5 = 250$ images from each method to evaluate. The same procedure is applied in the evaluation presented in Section 4.2. We asked 10 users from diverse backgrounds to evaluate the generated results in terms of text fidelity, content leakages and style similarity, and to provide their overall preference considering these three aspects. Finally, the final average results are displayed in Table 1 and Table 2.

## A.5 COEFFICIENT-TUNING RESULTS

We present the coefficient-tuning results of the IP-Adapter and InstantStyle in Figure 10. For IP-Adapter and InstantStyle, lowering the coefficient for image condition helps to enhance the control ability of text prompts, but it also comes with style degradation. Moreover, the coefficient-tuning process is quite labour-intensive. In contrast, our method can achieve much more stable results across various coefficient values. Figure 10 highlights the limitations of the InstantStyle approach in decoupling content and style, particularly regarding labour-intensive coefficient tuning.

Table 4: Evaluation Datasets in Section 4.1.

| | | Content | | | | | |
|---|---|---|---|---|---|---|---|
| **Train** | **Style References** | An automobile | An airplane | A bird | A cat | A deer | A dog |
| | | A horse | A frog | A ship | A truck | | |
| | | Styles | | | | | |
| | | natural environment | rainy day | snowy day | 3D model | abstract | cyberpunk |
| | | oil painting | realism | watercolor | beautiful landscape | watercolor | |
| | **Text Prompts** | Image reconstruction: use the content of style reference | | | | | |
| **Test** | **Style References** | Content | | | | | |
| | | An automobile | An airplane | A bird | A cat | A deer | A dog |
| | | A horse | A frog | A ship | A truck | | |
| | | Styles | | | | | |
| | | lego toy | statue | steampunk | stick figure | tattoo | line art |
| | | wool toy | surround with colored smoke | pop art | surround with white smoke | | |
| | **Text Prompts** | A bench | A human | A laptop | A rocket | A flower | |

## A.6 MORE VISUALIZATION RESULTS FOR SECTION 4

We present more visualization results for Section 4. In Figure 12-16, we mark the results with significant content leakages, style degradation, and loss of text fidelity with red, green, and blue boxes, respectively.

## A.7 COMPARISON BETWEEN OUR MASKING STRATEGY AND INSTANTSTYLE

To comprehensively demonstrate the effectiveness of our method in image-driven style transfer, we compare InstantStyle's feature subtraction (i.e., directly subtracting the content text feature from the image feature) and our masking-based method based on StyleShot's style encoder, providing visualization results in Figure 17 (a). Due to image-text misalignment, InstantStyle may disrupt the style information extracted by StyleShot's style encoder. In contrast, guided by appropriately-selected fewer conditions and without introducing additional content text features during the denoising process, our method successfully preserves the style features. Furthermore, we conduct ablation studies on the backbone of the diffusion model, as shown in Figure 17 (b). Using the SDXL diffusion model (Podell et al., 2023), we compare our method with InstantStyle's block-specific injection technique, which injects image features into style-specific blocks in the text-driven style transfer task. While InstantStyle encounters style disruption, our method significantly alleviates this problem by leveraging fewer appropriately-selected conditions.

## A.8 ABLATION STUDIES ON CLUSTERING NUMBER

We ablate on cluster number in the text-driven style transfer based on the StyleBench dataset. We report the image alignment and text alignment results based on three different CLIP backbones in Table 5 and provide the visual comparisons of our masking-based method with varying cluster numbers in Figure 18. It is shown that a smaller $K$, such as $K = 2$, can lead to better performance in avoiding content leakage, as more content-related elements in the style reference are masked. This is particularly evident in styles such as 3D models, Anime, and Baroque art, which contain more human-related images. In these cases, a smaller $K$ results in higher text alignment scores and more effective avoidance of content leakage.

## A.9 SIMULATION RESULTS OF PROPOSITION 1

We conduct simulation experiments based on our constructed dataset to demonstrate Proposition 1. Using the energy score proposed by (Liu et al., 2021), we calculate the energy scores of the masked image features for two different masking approaches: one based on clustering the product of $e_1^i$ and $e_2^i$ ($e_1^i \cdot e_2^i, i \in \{1, \cdots d\}$) and the other based on clustering the absolute difference of $e_1^i$ and $e_2^i$ ($|e_1^i - e_2^i|, i \in \{1, \cdots d\}$). For both methods, we report the 0th, 25th, 50th, 75th, and 100th percentiles of the energy scores. As shown in Table 6, our method consistently generates higher energy scores when discriminating content $c_2$, confirming the results outlined in Proposition 1.

## A.10 VISUAL COMPARISON IN TRADITIONAL STYLE TRANSFER

In Figure 21, we provide additional results comparing the performance of our method with previous approaches in traditional style transfer, where non-object style images are used as style references.

Table 5: Ablation study results on clustering number $K$.

| CLIP Model $K$ | ViT-B/32 | ViT-L/14 | ViT-H/14 |
|---|---|---|---|
| image alignment | | | |
| 2 | 0.657 | 0.608 | 0.403 |
| 3 | 0.656 | 0.611 | 0.410 |
| 4 | 0.657 | 0.615 | 0.415 |
| 5 | 0.657 | 0.614 | 0.415 |
| text alignment | | | |
| 2 | 0.265 | 0.212 | 0.258 |
| 3 | 0.264 | 0.211 | 0.253 |
| 4 | 0.265 | 0.210 | 0.252 |
| 5 | 0.264 | 0.210 | 0.252 |

| Style $K$ | 3D Model | Anime | Baroque |
|---|---|---|---|
| image alignment based on ViT-H/14 | | | |
| 2 | 0.474 | 0.372 | 0.384 |
| 3 | 0.478 | 0.381 | 0.393 |
| 4 | 0.485 | 0.390 | 0.404 |
| 5 | 0.487 | 0.380 | 0.411 |
| text alignment based on ViT-H/14 | | | |
| 2 | 0.213 | 0.234 | 0.257 |
| 3 | 0.206 | 0.232 | 0.253 |
| 4 | 0.189 | 0.231 | 0.253 |
| 5 | 0.188 | 0.229 | 0.252 |

Table 6: The energy scores of the masked image features for two different masking approaches. We report the 0th, 25th, 50th, 75th, and 100th percentiles of the energy scores for each approach.

| Masking Proportion | Method | 0th | 25th | 50th | 75th | 100th |
|---|---|---|---|---|---|---|
| 5% | clustering on $e_1^i \cdot e_2^i$ (Ours) | -10.78 | -6.59 | -5.36 | -4.08 | 1.64 |
| 5% | clustering on $|e_1^i - e_2^i|$ | -13.87 | -9.63 | -8.56 | -7.54 | -1.60 |
| 10% | clustering on $e_1^i \cdot e_2^i$ (Ours) | -9.15 | -5.15 | -4.00 | -2.77 | 2.02 |
| 10% | clustering on $|e_1^i - e_2^i|$ | -11.57 | -8.80 | -7.88 | -7.05 | -2.18 |
| 20% | clustering on $e_1^i \cdot e_2^i$ (Ours) | -7.46 | -3.46 | -2.36 | -1.35 | 2.66 |
| 20% | clustering on $|e_1^i - e_2^i|$ | -10.73 | -7.57 | -6.91 | -6.20 | -3.16 |
| 30% | clustering on $e_1^i \cdot e_2^i$ (Ours) | -6.05 | -2.62 | -1.58 | -0.63 | 2.87 |
| 30% | clustering on $|e_1^i - e_2^i|$ | -9.19 | -6.73 | -6.13 | -5.59 | -3.31 |
| 40% | clustering on $e_1^i \cdot e_2^i$ (Ours) | -5.71 | -2.23 | -1.29 | -0.43 | 2.70 |
| 40% | clustering on $|e_1^i - e_2^i|$ | -8.04 | -5.99 | -5.51 | -5.07 | -3.61 |
| 50% | clustering on $e_1^i \cdot e_2^i$ (Ours) | -5.24 | -1.94 | -1.13 | -0.37 | 2.69 |
| 50% | clustering on $|e_1^i - e_2^i|$ | -7.26 | -5.37 | -4.99 | -4.60 | -3.63 |
| 60% | clustering on $e_1^i \cdot e_2^i$ (Ours) | -4.93 | -1.72 | -0.92 | -0.22 | 2.72 |
| 60% | clustering on $|e_1^i - e_2^i|$ | -6.23 | -4.85 | -4.53 | -4.19 | -3.29 |
| 70% | clustering on $e_1^i \cdot e_2^i$ (Ours) | -3.91 | -1.25 | -0.53 | 0.15 | 2.86 |
| 70% | clustering on $|e_1^i - e_2^i|$ | -5.62 | -4.32 | -4.06 | -3.77 | -2.98 |
| 80% | clustering on $e_1^i \cdot e_2^i$ (Ours) | -3.11 | -0.77 | -0.14 | 0.53 | 2.20 |
| 80% | clustering on $|e_1^i - e_2^i|$ | -4.72 | -3.77 | -3.57 | -3.36 | -2.57 |
| 90% | clustering on $e_1^i \cdot e_2^i$ (Ours) | -2.40 | -0.67 | -0.18 | 0.37 | 2.12 |
| 90% | clustering on $|e_1^i - e_2^i|$ | -4.00 | -3.32 | -3.15 | -3.01 | -2.55 |

As shown in Figure 21, when using non-object style references, previous methods such as StyleDrop (Sohn et al., 2024) and StyleShot (Gao et al., 2024) may suffer from style degradation or a loss of text fidelity. As pointed out in the original paper (Rout et al., 2024), "The inherent limitations of the style descriptor or diffusion model might propagate into our framework", RB-Modulation Rout et al. (2024) may fail to preserve the style of the reference when the style description does not align well with the image reference, as illustrated in the 3rd and 4th lines in Figure 21. As shown in the 9th and 10th lines in Figure 21, CSGO (Xing et al., 2024) may also suffer from style degradation or loss of text fidelity, showing inferior performance compared to our method. In contrast, from Figure 6 and Figure 21, the proposed method demonstrates superior stylization, performing better in both object-centered style references and non-object style references.

## A.11 ETHICS STATEMENT

This work aims to make a positive impact on the field of AI-driven image generation. We aim to facilitate the creation of images with diverse styles, and we expect all related processes to comply with laws and be used responsibly.

The use of AI to generate human-related images, particularly those involving characteristics such as skin color, gender, age, and other demographic factors, raises complex ethical questions. We are aware that the generation of images involving these attributes must be handled with care to avoid reinforcing stereotypes, perpetuating discrimination, or contributing to the misrepresentations of certain groups. We take these concerns very seriously and believe that AI should be used in a way that promotes fairness, inclusion, and respect for all individuals. Here, we give several examples of text prompts containing different genders, skin colors, and ages, as shown in Figure 22. We observe that in most cases, our method is able to generate images with diversity. However, there are certain cases that general image-generation methods can be misused, and caution must be exercised to mitigate any potential harm.

Our methodology builds upon existing open-source diffusion models (e.g., Stable Diffusion V1.5 (Rombach et al., 2022) and SDXL (Podell et al., 2023)), which already include some built-in safeguards, such as the NSFW ("not safe for work") Filter. We believe that our contributions in generating images with diverse styles, while preventing content leakages, can further positively help avoid stereotypes and discriminations. Moreover, successfully preventing the leakage of style reference' content into the generated image can avoid the unintended blending of the target subject with the style reference' content, thereby helping to mitigate malicious or offensive generations. Furthermore, we are committed to ongoing research in this area, aiming to enhance safety mechanisms and prevent the creation of malicious or offensive imagery. Ultimately, we believe that the responsible use of AI-driven image-generation tools is essential to fostering ethical and equitable outcomes in the field.

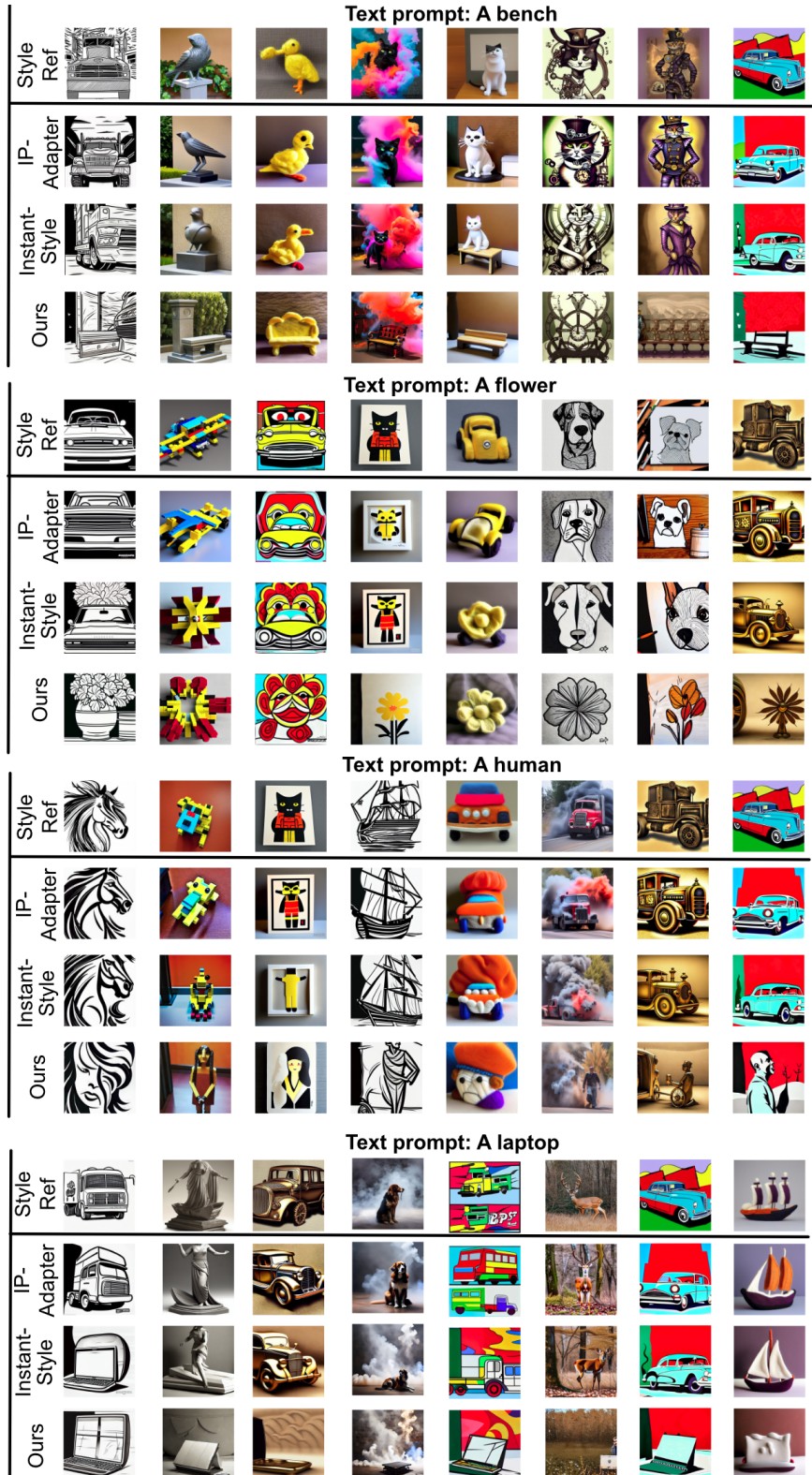

Figure 9: **Effectiveness of the proposed masking strategy over feature subtraction in avoiding content leakages.** The superior performance of our method showcases that the appropriately-selected fewer conditions can more efficiently avoid content leakages.

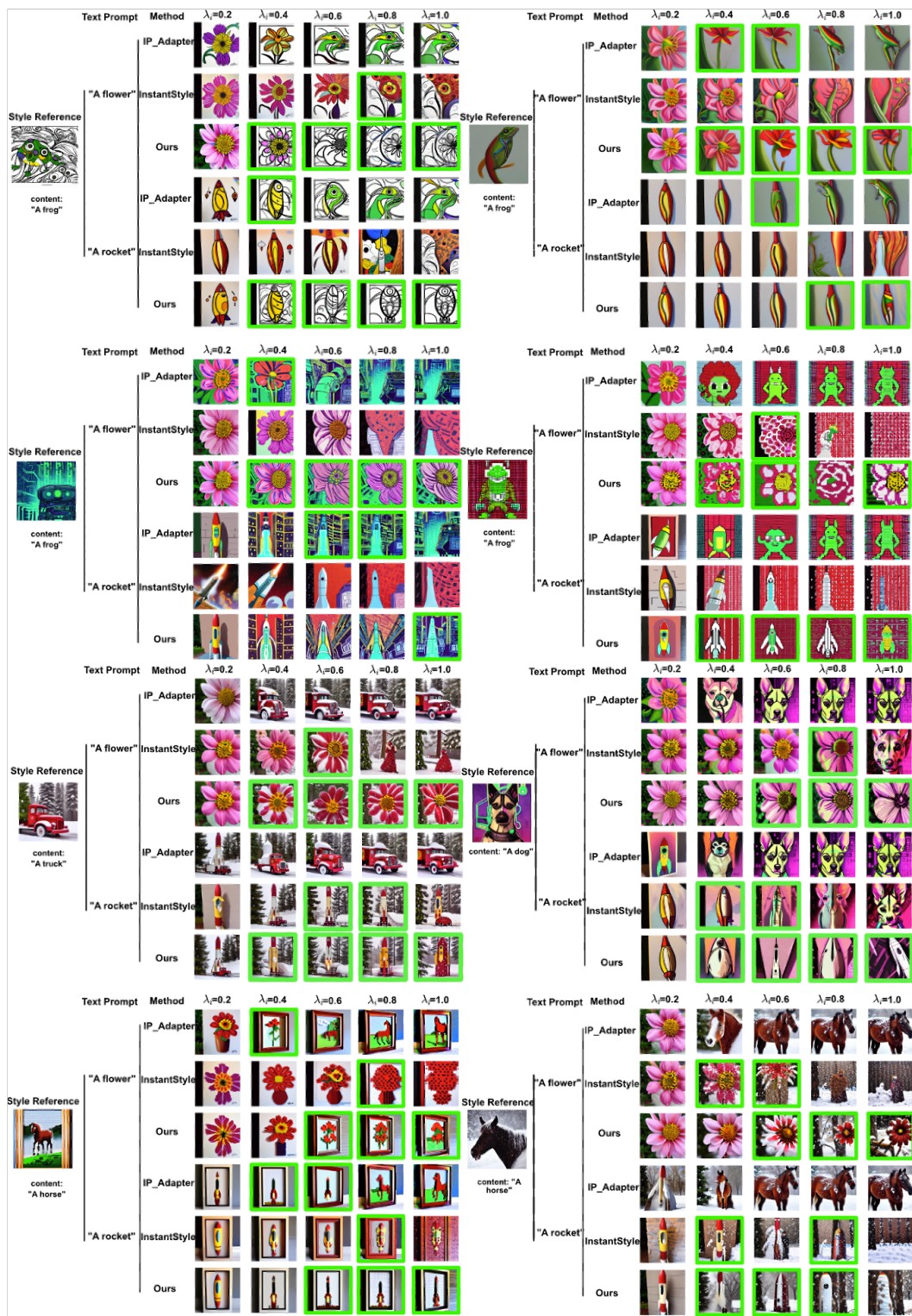

Figure 10: **The coefficient-tuning results of IP-Adapter (Ye et al., 2023), InstantStyle (Wang et al., 2024a) and our proposed method.** We use different coefficients for the image condition (i.e., $\lambda_i$ in Equation 2). We highlight the satisfactory results with green boxes.

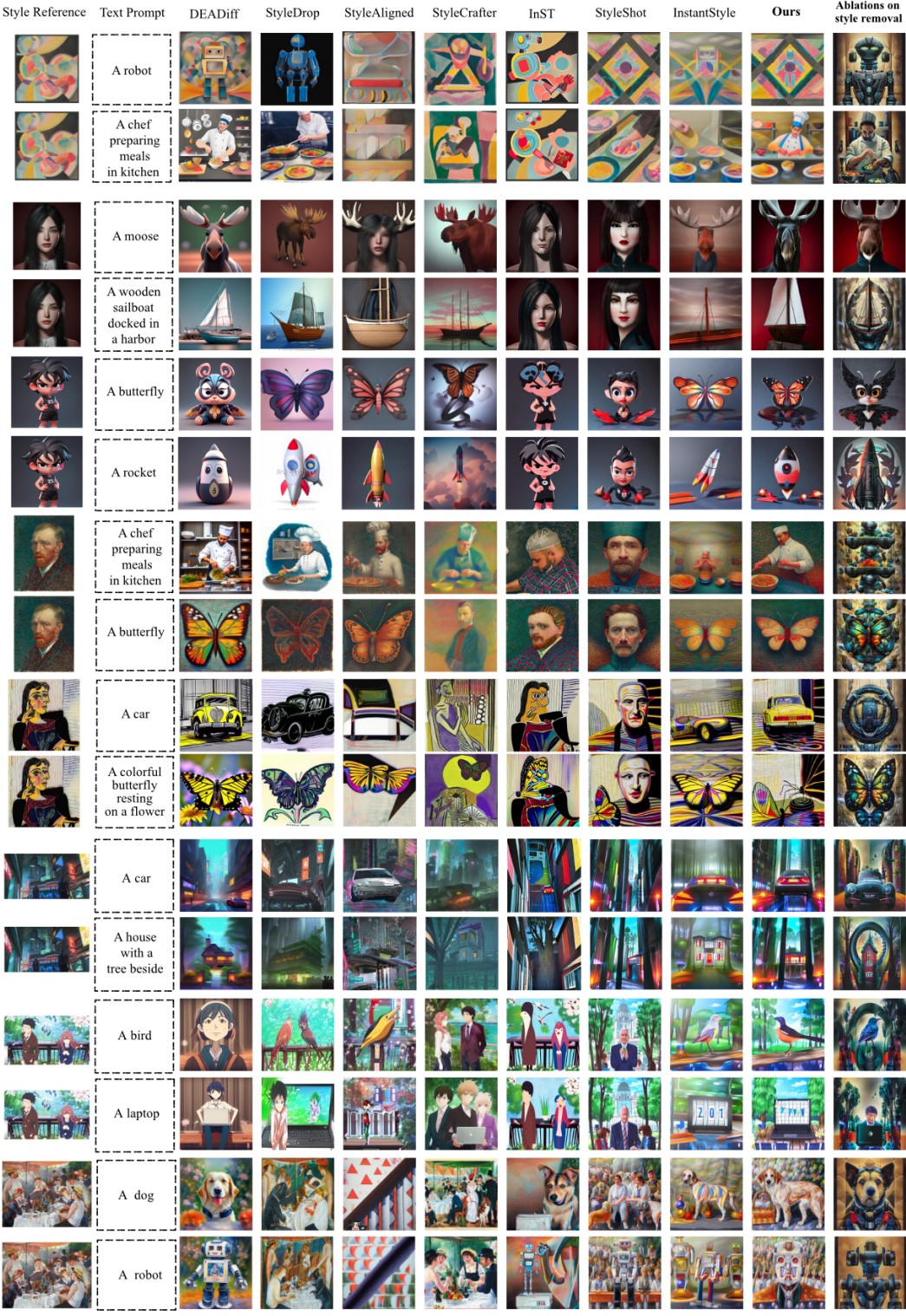

Figure 11: **Visual comparison between recent state-of-the-art methods and ours in text-driven style transfer on StyleBench.** The proposed masking-based method does not require content knowledge of the image reference; instead, we leverage the CLIP text feature of "person, animal, plant, or object in the foreground" to identify the elements that need to be masked.

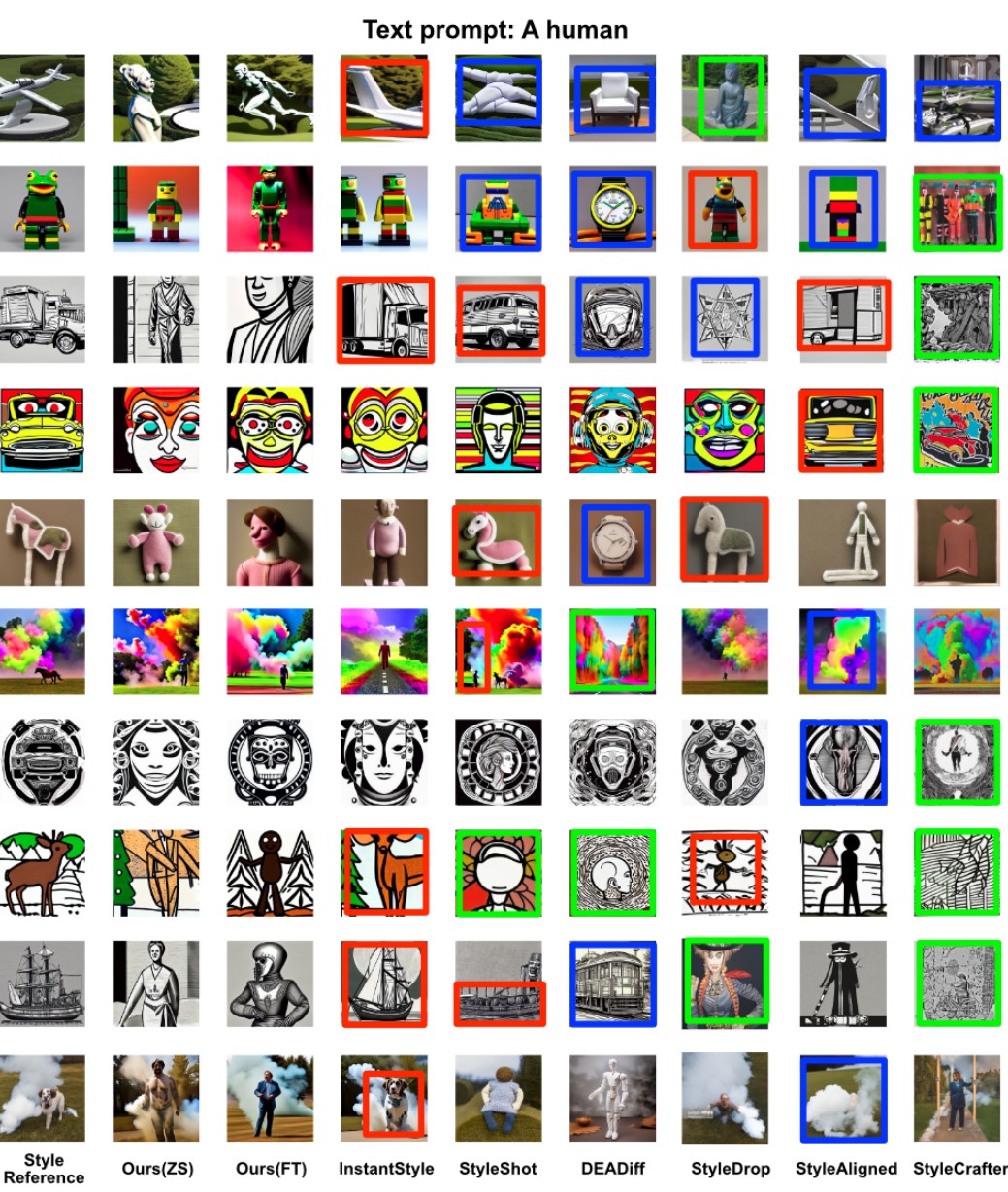

Figure 12: Visual comparison between recent state-of-the-art methods and ours in text-driven style transfer, where the text prompt is "A human".

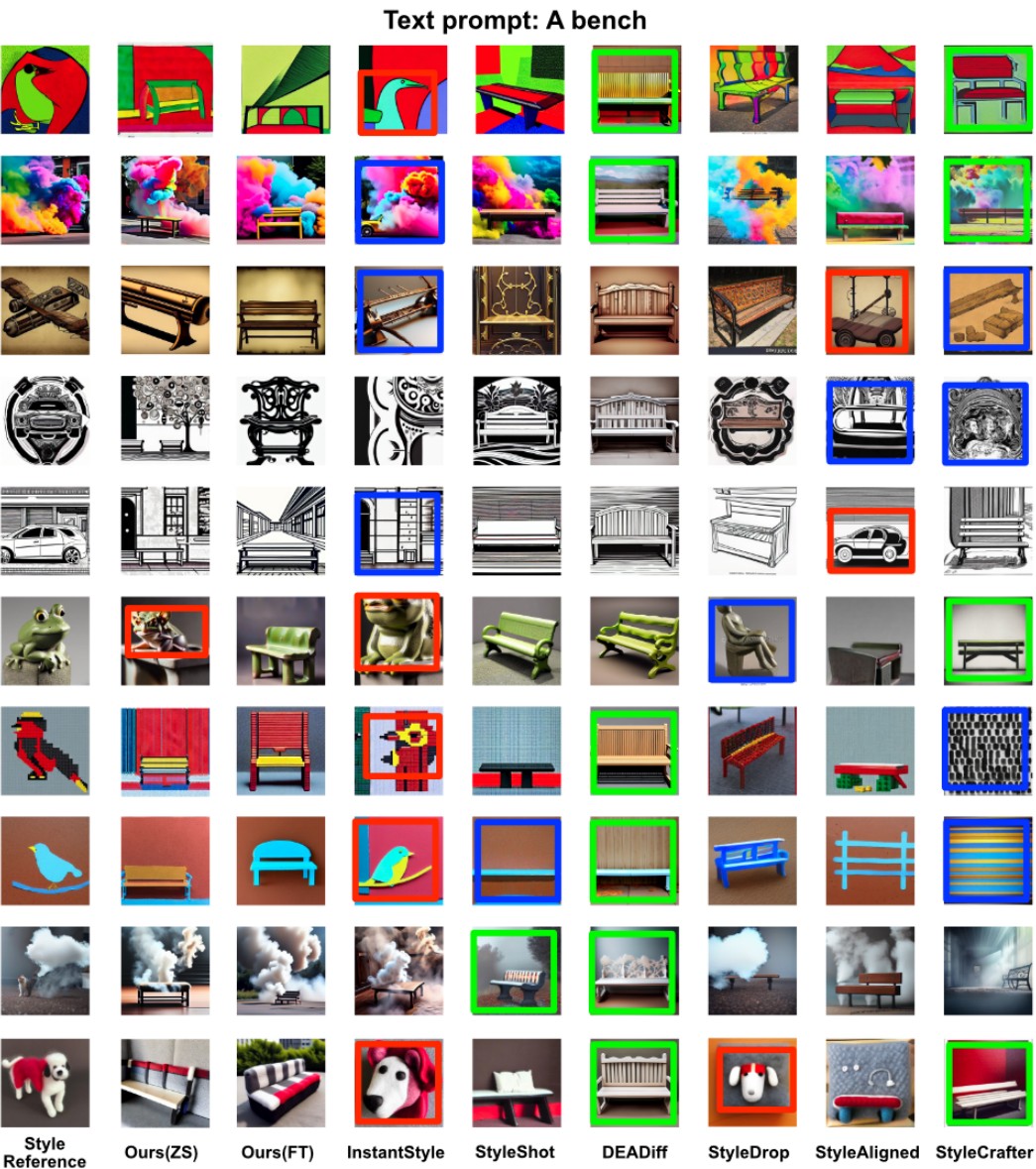

Figure 13: Visual comparison between recent state-of-the-art methods and ours in text-driven style transfer, where the text prompt is "A bench".

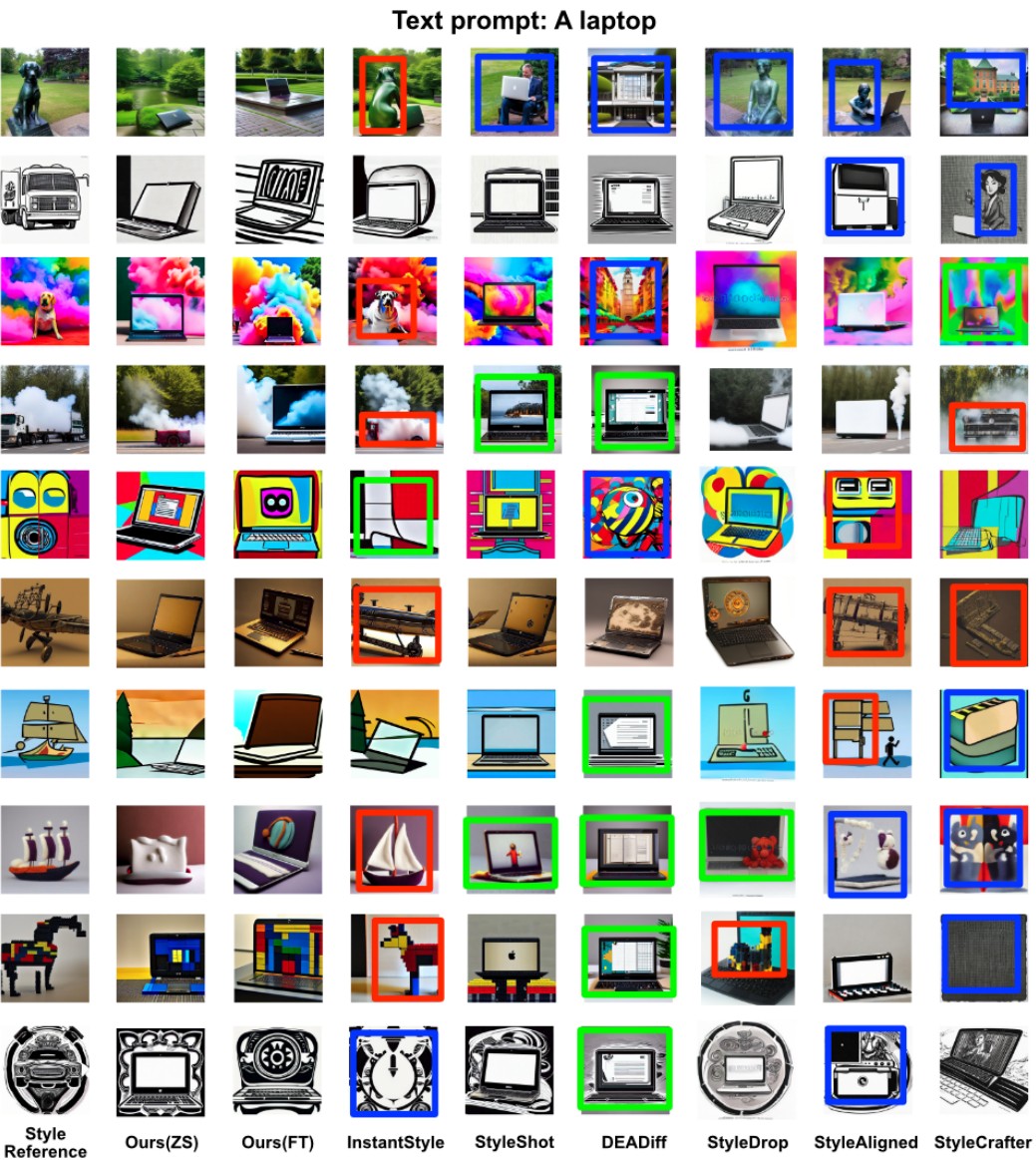

Figure 14: Visual comparison between recent state-of-the-art methods and ours in text-driven style transfer, where the text prompt is "A laptop".

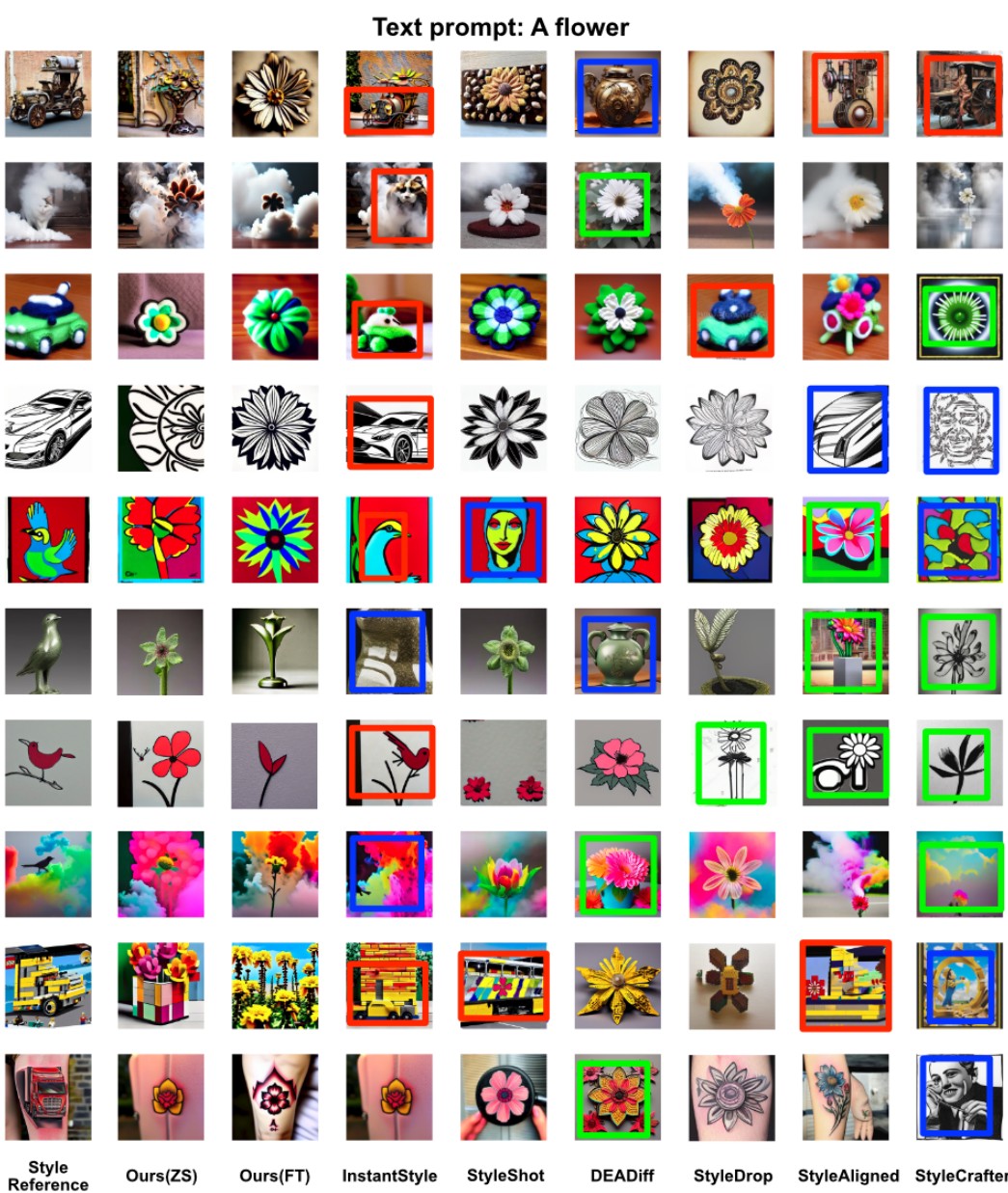

Figure 15: Visual comparison between recent state-of-the-art methods and ours in text-driven style transfer, where the text prompt is "A flower".

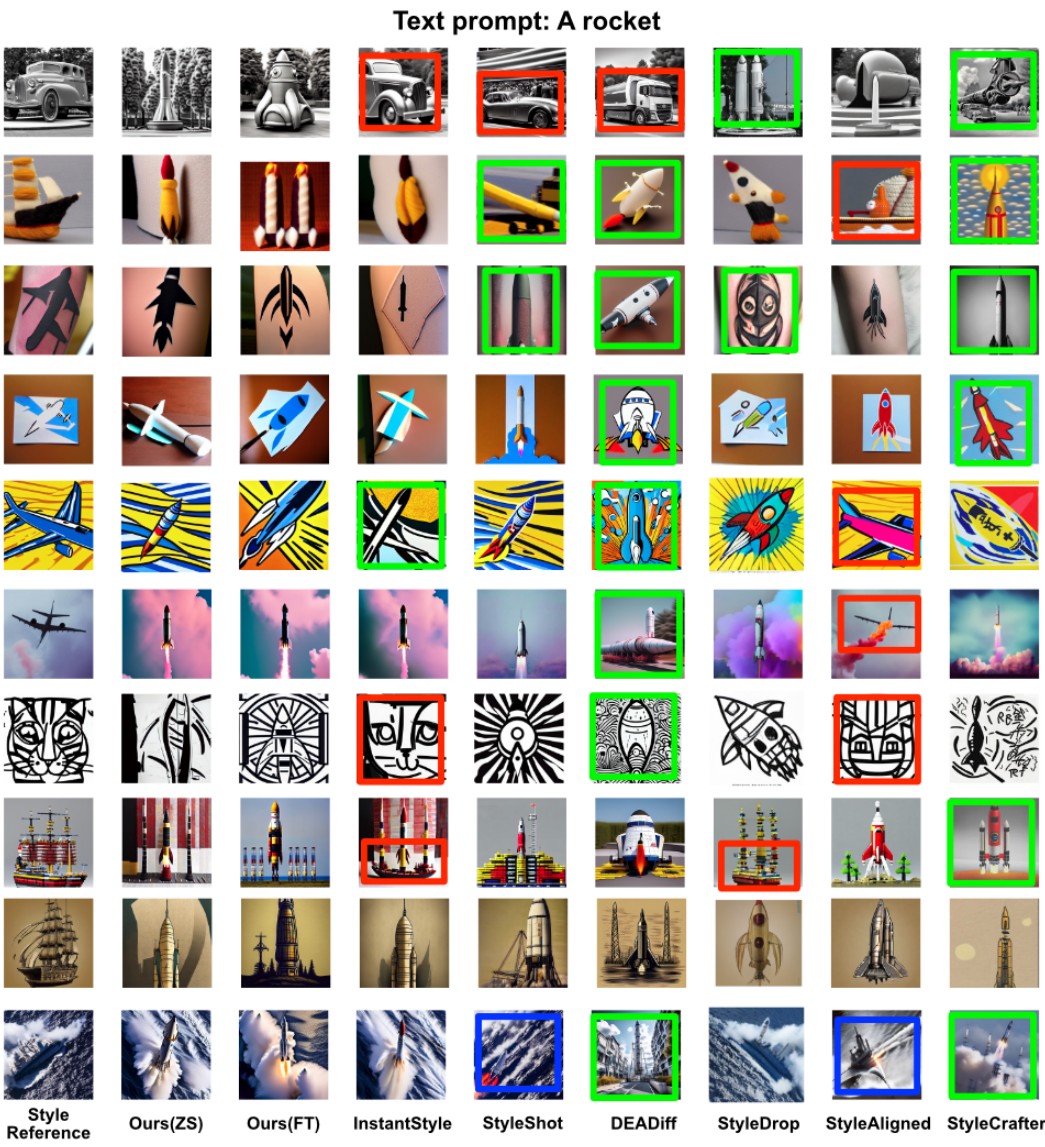

Figure 16: Visual comparison between recent state-of-the-art methods and ours in text-driven style transfer, where the text prompt is "A rocket".

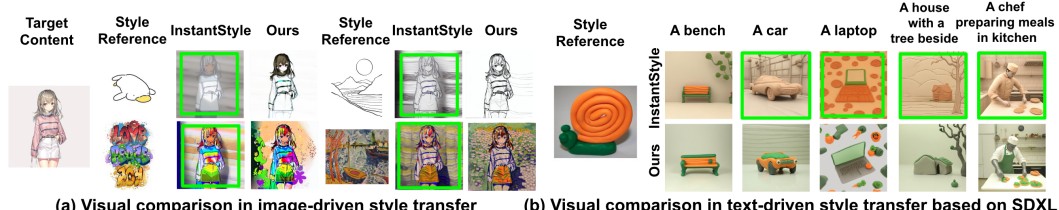

Figure 17: (a) Visual comparison between InstantStyle's feature subtraction and ours in image-driven style transfer. (b) Visual comparison between InstantStyle's block-specific injection techniques and ours in text-driven style transfer based on SDXL.

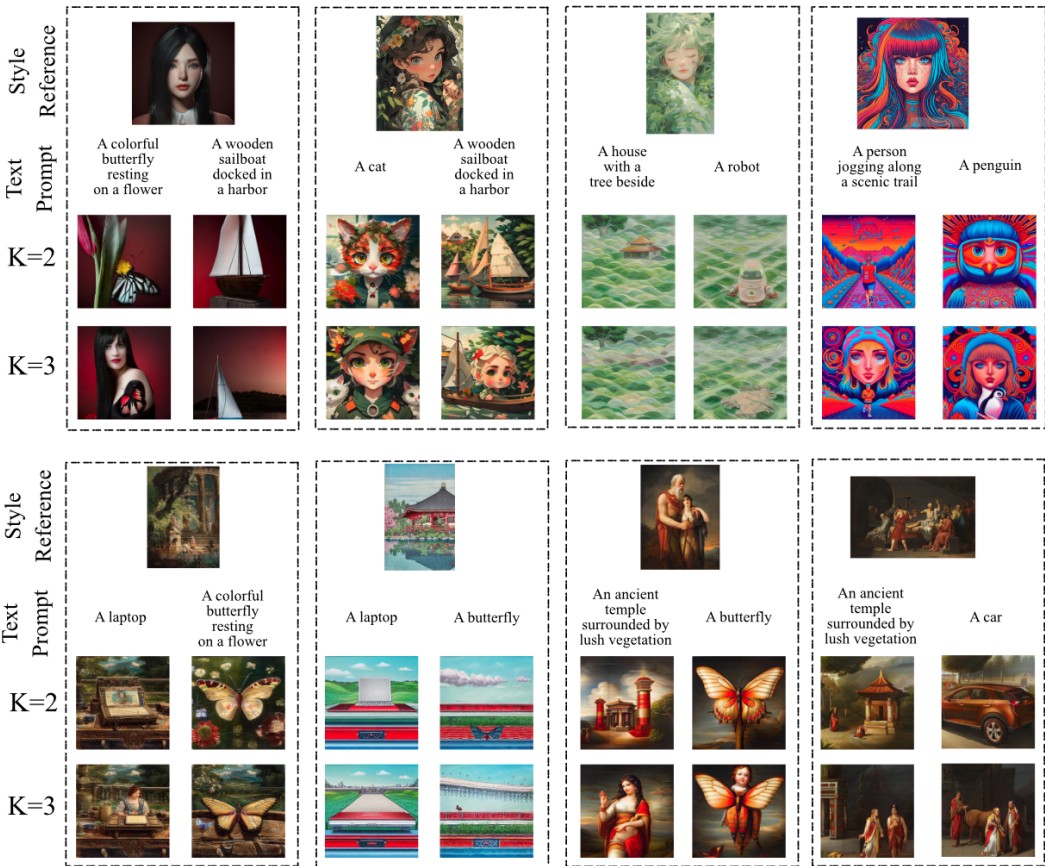

Figure 18: Visual comparison of the proposed masking-based method with varying cluster numbers. It is shown that a smaller $K$, such as $K = 2$, can lead to better performance in avoiding content leakage, as more content-related elements in the style reference are masked. This is particularly evident in styles such as 3D models, Anime, and Baroque art, which contain more human-related images. In these cases, a smaller $K$ results in higher text alignment scores and more effective avoidance of content leakage.

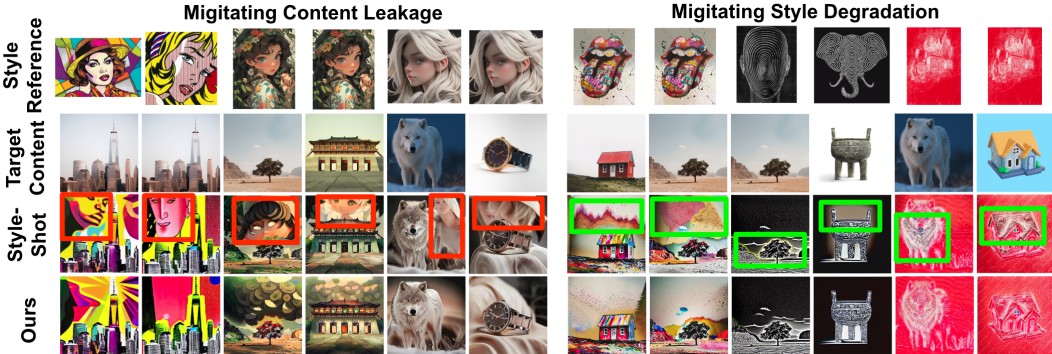

Figure 19: Visual comparison between StyleShot and ours in image-driven style transfer. Results with content leakages and style degradation are highlighted with red and green boxes, respectively.

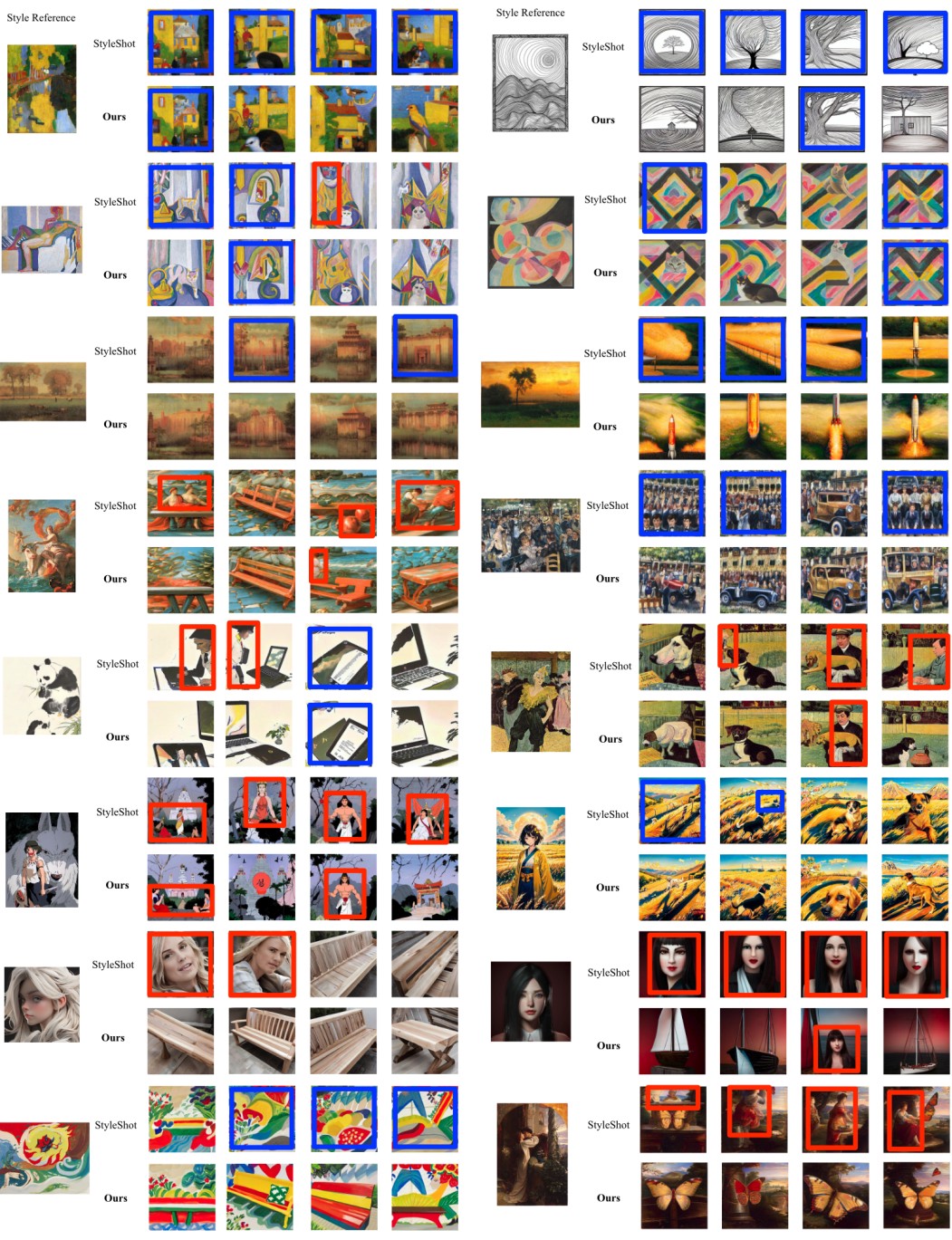

Figure 20: The one-to-one image generation results between the proposed masking-based method and StyleShot. We provide multi-sample generalization results for each combination of style reference and target text prompt. To mitigate the influence of randomness, we ensure that all model configurations remain consistent, including the random seed, guidance seed, denoising steps, and other parameters. From the one-to-one comparison, we observe that our method significantly reduces content leakage and alleviates loss of text fidelity, consistently refining StyleShot's results across all combinations. We mark the results with significant content leakages and loss of text fidelity with red and blue boxes, respectively.

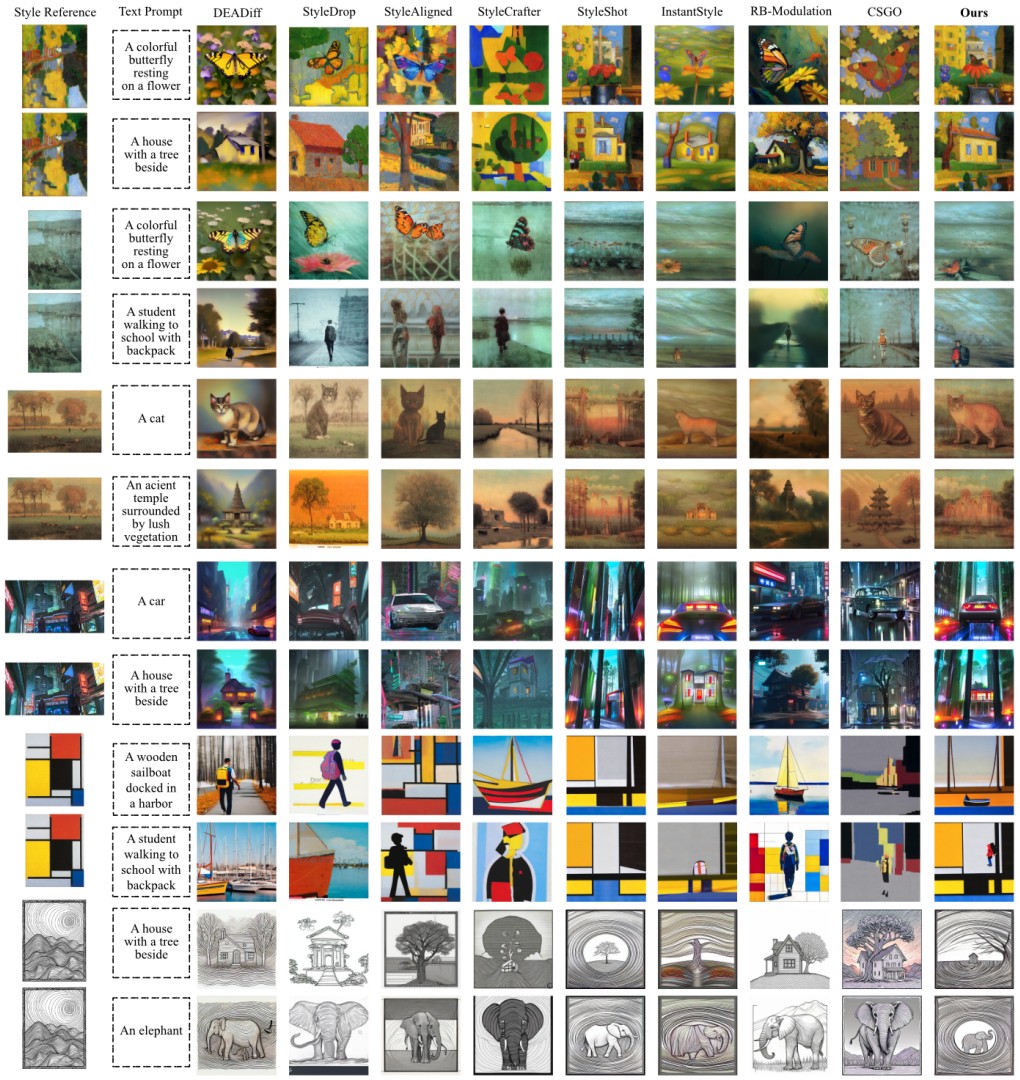

Figure 21: Visual comparison between recent state-of-the-art methods in traditional style transfer.

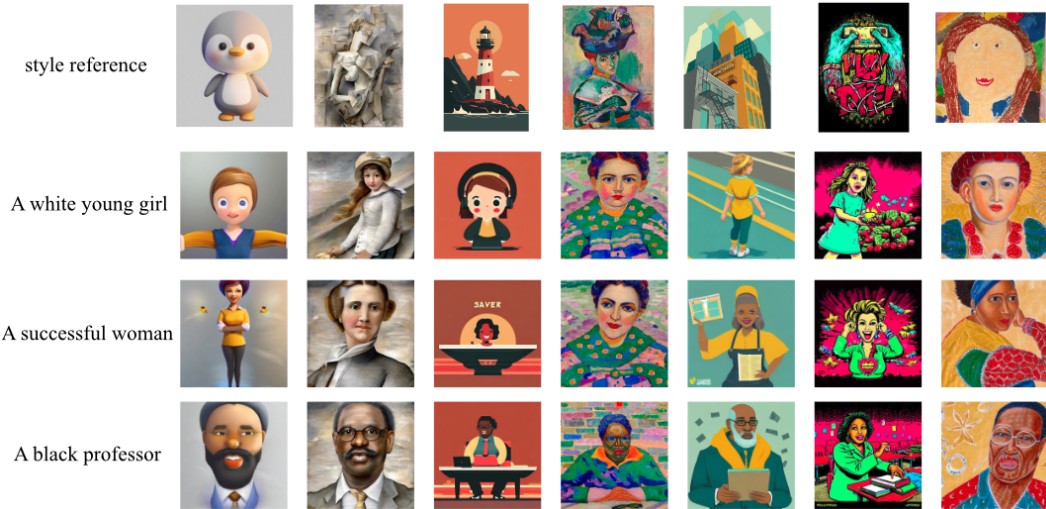

Figure 22: Examples of text prompts contain different genders, skin colors, and ages.

