# OpenReview forum: "Less is More: Masking Elements in Image Condition Features Avoids Content Leakages in Style Transfer Diffusion Models"
_ICLR.cc/2025/Conference — ICLR 2025 Poster_

### Official Review · Reviewer_nfUg · 2024-10-29

**Soundness:** 3
**Presentation:** 3
**Contribution:** 3
**Rating:** 8
**Confidence:** 4

**Summary:**

The paper proposes a feature masking method to control the content leakage for the stylization task. It requires an additional content description of the image and decouples this content by masking it out in the style features. The authors also provided theoretical proofs to demonstrate the motivation of their method. Both qualitative and quantitative results are superior to the alternatives.

**Strengths:**

1. The proposed method is intuitive and works well.
2. Theoretical proofs are valid and well support the experiments.
3. Generally the writing is good and the story is complete, though there is some information missing as I mentioned in the following section.
4. Experiment design is comprehensive and the performance looks good.

**Weaknesses:**

1. The authors should compare with more recent stronger baselines that are proposed to alleviate the content leakage problem like RB-Modulation, which uses attention feature aggregation and different descriptors to decouple content and style. Since it is also training-free and mentioned to outperform InstantStyle, it would serve as a good baseline for comparison. CSGO is another recent work that uses a separately trained style projection layer to avoid content leakage. Though it’s pretty new (released a month before deadline), some qualitative results would help demonstrate the strengths of your method.
2. In L229, should m^i be 1 or 0?
3. Figure 3(b) is not referred to but seems to be mentioned in the experiments. I thought this should be part of the method you want to introduce. Can you explain how this is used with your proposed method? And how does the linear layer learn the content feature to be subtracted?
4. Theorem 1 indicates that the proposed method archives a smaller divergence. Does “smaller divergence” define better style alignment? I’m asking because there might be several factors that can lead to smaller divergence, like same background/or elements in the images, content leakage, etc. I’d like to get your insights on what is the style in an image?
5. In L215, it seems cluster number K controls how many tokens are masked. Is there any analysis to show how K affects the performance?
6. Can you add more details on how you do binary classification as eval in L369?
7. Are you using style descriptions in the prompt?
8. Why does image alignment keep dropping in figure 5(a)?
9. Can you provide more details on how you conduct user study? Like instructions to the raters and how you present the images to the raters.

The answers can be kept short and concise since there are many questions. Thanks!

**Questions:**

See weakness

---

> ### Author Response · Authors · 2024-11-23
> **Part 1/2**
>
> **Reply to weakness 1:**
>
> We provide visual comparisons between our method, RB-Modulation, and CSGO. The results can be accessed at the following links:  https://drive.google.com/file/d/1jjAgViV9Z7MTrES0eMhaBPlpS2ZtyEVE/view?usp=sharing and https://drive.google.com/file/d/1zQQa2XHlTBspOJ1IvC6VKQnRxEuu2Lvc/view?usp=drive_link. We present several key observations:
>
>    1. The CSGO method may suffer from style degradation or loss of text fidelity, showing inferior performance compared to our method.
>    2. The RB-Modulation method proposes a framework that incorporates a style descriptor into a pre-trained diffusion model. With style descriptions, RB-Modulation can generate more satisfactory results than without them, preventing information leakage from the reference style and adhering more closely to the desired prompt.
>    3. As pointed out in the original paper, ''The inherent limitations of the style descriptor or diffusion model might propagate into our framework''. Thus, RB-Modulation may fail to preserve the style of the reference when the style description does not align well with the image reference. For example, RB-Modulation’s results may experience style degradation on abstract, Orphism, or realism art styles."
>    4. Building upon the impressive StyleShot and leveraging appropriately fewer conditions, our method successfully avoids content leakage while enhancing style, achieving better style transfer performance than recent competitive models.
>
> **Reply to weakness 2:**
>
> Sorry for the confusion; m^i should be set to 0 to discard the content-related elements.
>
> **Reply to weakness 3:**
>
> We apologize for the confusion. Figure 3(b) illustrates the Learning Paradigms introduced in Lines 305-318. In the Learning Paradigms, the style reference's image feature $e_1$ is subtracted by the content feature $\psi(e_1)$ or $\phi(e_2)$ to avoid the presence of content $c_2$. We will add an explanation for clarity. As pointed out in Line 361, the linear layers are trained through image reconstruction using mean squared error (MSE) loss to predict errors. The specific algorithm is provided in Algorithm 2 of Appendix A.3. The optimization objectives are illustrated in Lines 877 and 879 for the Text-Adapter (Baseline) and Image-Adapter (Ours), respectively. These optimization objectives are used to minimize the prediction error of noise in reconstructing style reference while maximizing the difference between the model conditioned on the style reference's content and that conditioned on the target prompt.
>
> **Reply to weakness 4:**
>
> The ground-truth condition data distribution ($q(x|c_1, c_2, c_3)$) defines images that exhibit high style similarity with the style reference while avoiding content from the style reference while maintaining high fidelity with the target text prompt. Thus, "smaller divergence" indicates that the result distribution generated by our proposed method is closer to the ground-truth distribution, showing better style alignment, text fidelity, or less content leakage.

---

> ### Author Response · Authors · 2024-11-23
> **Part 2/2**
>
> **Reply to weakness 5:**
>
> We ablate on cluster number $K$ in the text-driven style transfer based on the StyleBench dataset. We report the image alignment and text alignment results based on three different CLIP backbones in the following Table. It is shown that a smaller $K$, such as $K=2$ can lead to a slightly higher text alignment score since more content-related elements in the style reference are masked. Especially in the 3D model, Anime, and Baroque art style that contains more human-related images,  smaller $K$ can lead to higher text alignment scores and more efficiently avoiding content leakage.
>
>    |                 | ViT-B/32 |      | ViT-L/14 |      | ViT-H/14 |
>    | --------------- | -------- | ---- | ----- | ---- | -------- |
>    | Image Alignment |          |      |       |      |          |
>    | K=2             | 0.657    | K=2  | 0.608 | K=2  | 0.403    |
>    | K=3             | 0.656    | K=3  | 0.611 | K=3  | 0.410    |
>    | K=4             | 0.657    | K=4  | 0.615 | K=4  | 0.415    |
>    | K=5             | 0.657    | K=5  | 0.614 | K=5  | 0.415    |
>    | Text Alignment  |          |      |       |      |          |
>    | K=2             | 0.265    | K=2  | 0.212 | K=2  | 0.258    |
>    | K=3             | 0.264    | K=3  | 0.211 | K=3  | 0.253    |
>    | K=4             | 0.265    | K=4  | 0.210 | K=4  | 0.252    |
>    | K=5             | 0.264    | K=5  | 0.210 | K=5  | 0.252    |
>
>    |                                 | 3D Model |      | Anime |      | Baroque |
>    | ------------------------------- | -------- | ---- | ----- | ---- | ------- |
>    | Image  Alignment based on ViT-H/14 |          |      |       |      |         |
>    | K=2                             | 0.474    | K=2  | 0.372 | K=2  | 0.384   |
>    | K=3                             | 0.478    | K=3  | 0.381 | K=3  | 0.393   |
>    | K=4                             | 0.485    | K=4  | 0.390 | K=4  | 0.404   |
>    | K=5                             | 0.487    | K=5  | 0.380 | K=5  | 0.411   |
>    | Text  Alignment based on ViT-H/14  |          |      |       |      |         |
>    | K=2                             | 0.213    | K=2  | 0.234 | K=2  | 0.257   |
>    | K=3                             | 0.206    | K=3  | 0.232 | K=3  | 0.253   |
>    | K=4                             | 0.189    | K=4  | 0.231 | K=4  | 0.253   |
>    | K=5                             | 0.188    | K=5  | 0.229 | K=5  | 0.252   |
>
> **Reply to weakness 6:**
>
> We apologize for the confusion. We detailed this in Line 913. We perform binary classification on the generated images to differentiate between the reference's content object and the text prompt, computing the classification accuracy, which is referred to as the fidelity score. Therefore, the fidelity score primarily reflects the model’s ability to control text prompts.
>
> **Reply to weakness 7:**
>
> In all experiments, we **do not** use any style descriptions in the prompt. The only input difference between our method and the baseline model, IP-Adapter, is the content description for the style reference, which we simply use a common template: "person, animal, plant, or object in the foreground" in experiments on the StyleBench benchmark. That is, the proposed masking-based method does not require content knowledge of the image reference; instead, we leverage the CLIP text feature of a common template to identify the elements that need to be masked.
>
> **Reply to weakness 8:**
>
> Following the metric used in Gao et al., we report the image and text alignment scores alongside training steps in Figure 5(a). Image alignment refers to the cosine similarity between the CLIP embeddings of the generated images and the style reference images, while text alignment measures the cosine similarity between the CLIP embeddings of the generated images and the target text prompts.
>
> Initially, the image alignment is high due to significant content leakage in the generated image. As the unwanted content decreases, the alignment between the generated image and the style reference gradually decreases.
>
>   [1] Junyao Gao, Yanchen Liu, Yanan Sun, Yinhao Tang, Yanhong Zeng, Kai Chen, and Cairong Zhao. Styleshot: A snapshot on any style. arXiv preprint arXiv:2407.01414, 2024.
>
> **Reply to weakness 9:**
>
> We asked 10 users from diverse backgrounds to evaluate the generated results in terms of text fidelity, content leakage, and style similarity, and to provide their overall preference for each composition of style reference and target text prompt, considering these three aspects. We sampled 50 generated images for each target text prompt and provided all results to them.

---

> > ### Comment · Reviewer_nfUg · 2024-11-23
> >
> > It seems both attached links in the reponse to weakness 1 are the same, maybe a typo?

---

> > > ### Author Response · Authors · 2024-11-24
> > >
> > > We apologize for the typo. The links have been updated, and the results can now be accessed at:https://drive.google.com/file/d/1jjAgViV9Z7MTrES0eMhaBPlpS2ZtyEVE/view?usp=sharing and https://drive.google.com/file/d/1zQQa2XHlTBspOJ1IvC6VKQnRxEuu2Lvc/view?usp=drive_link.

---

> > > > ### Comment · Reviewer_nfUg · 2024-11-24
> > > >
> > > > re 1: thanks for the addtional comparison. Please add the discussion to the main draft in a new revision, since they are more relevant to the leakage problem discussed in the paper.
> > > >
> > > > re 2: please make the correction in a new reivision.
> > > >
> > > > re 3: add reference to the figure where needed in the new revision.
> > > >
> > > > re 5: This is an important ablation in the paper, make sure it is added to the paper.
> > > >
> > > > re 6: I saw L913. I was asking what details of the binrary classification. e.g., which model did you use for binary classification? any threshold used? etc.
> > > >
> > > > re 9: I saw the part you quote in the appendix. You have 10 content objects (from cifar10) and 21 image styles (L893), 50 generations for each prompt. so 21x50=1050 images evaluated for each class per rater, is this true?

---

> ### Author Response · Authors · 2024-11-25
>
> We sincerely thank you for your valuable comments and apologize for any confusion.
>
> We have carefully incorporated all of your suggestions into our revised manuscript and have submitted it for your review. We have revised both the main manuscript and the appendix, with all changes marked in red for your convenience.
>
> Further explanations are provided for replies 6 and 9.
>
> For re 6: The binary classification is performed using the CLIP model, with the CLIP-H/14 as the image encoder. Specifically, we denote the cosine similarity between the CLIP image feature of the generated image and the CLIP text feature of reference's content object as $\frac{\langle e_2, e_g \rangle}{|e_2| \cdot |e_g|}$. Similarly, we denote the cosine similarity between the CLIP image feature of the generated image and the CLIP text feature of text prompt as $\frac{\langle e_3, e_g \rangle}{|e_3| \cdot |e_g|}$. If $\frac{\langle e_2, e_g \rangle}{|e_2| \cdot |e_g|} < \frac{\langle e_3, e_g \rangle}{|e_3| \cdot |e_g|}$, the generated image is considered correctly classified, meaning it contains the target content rather than the content of the style reference.
>
>
> For re 9:
> Apologies for the confusion. In Section 4.1, the constructed dataset consists of 10 content objects (from CIFAR-10) and 21 image styles (11 for training and 10 for testing) for each content object, with 8 variations per style. This results in a total of $10 \times 11 \times 8 = 880$ style references. For each style reference, we perform style transfer for 5 target text prompts, with 4 generations per target text prompt, leading to $880 \times 4 = 3520$ generations per text prompt. We randomly sample 50 images from the 3520 generated images for each target text prompt. In total, this gives us $50 \times 5 = 250$ images from each method to evaluate. The same procedure is applied in the evaluation presented in Section 4.2.
> We have also included these details in Appendix A.4.2 for further clarity.

---

> > ### Comment · Reviewer_nfUg · 2024-11-25
> > **increase score**
> >
> > The authors addressed all my concerns, increased score. As pointed out by the authors, the proposed method do not rely on style descriptions to achieve good performance where most of exsiting state-of-the-art fail to do so.

---

> > > ### Author Response · Authors · 2024-11-26
> > > **Thanks for your valuable time and suggestions**
> > >
> > > Thank you for your recognition and support of our work. We sincerely appreciate the time and effort you have devoted to considering our response. Your constructive suggestions have greatly enhanced the quality of our manuscript. We will continue to refine and improve its presentation.

---

### Official Review · Reviewer_gkxk · 2024-10-31

**Soundness:** 3
**Presentation:** 3
**Contribution:** 3
**Rating:** 6
**Confidence:** 5

**Summary:**

This paper proposes a simple but effective method to avoid content leakages and achieve better performance in style transfer. The main contribution is its innovative masking strategy, and extensive experiments demonstrate its superiority.

**Strengths:**

1. The paper is well-written and easy to follow.
2. The proposed masking strategy is novel and effective.
3. The authors provide both theoretical and experimental evidence to support their claims.

**Weaknesses:**

1. It remains unclear why clustering is performed on the element-wise product of $e_1$ and $e_2$.  Is there a relationship between $e_1\cdot e_2$ and the energy function?
2. The inference speed is slower than other methods, likely due to the additional time consumption introduced by the clustering algorithm. What is your inference time in practice? Is there a solution that avoids this additional time cost, or could the clustering algorithm be replaced to improve efficiency?

**Questions:**

See Weaknesses.

---

> ### Author Response · Authors · 2024-11-23
>
> **Reply to Weakness1:**
>
> We perform clustering on the element-wise product of $e_1$ and $e_2$. Compared to other clustering methods, such as clustering on the element-wise absolute difference between $e_1$ and $e_2$, our method achieves the highest energy score for content $c_2$ after masking elements in the high-mean cluster, as shown in Proposition 1. The high value of the energy function $\mathcal{E}(c_2, x_t)$ indicates $x_t$ exhibits a low likelihood of content $c_2$, leading to superior performance in content removal.
>
>
> We conduct simulation experiments based on our constructed dataset (which we introduced in Sec 4.1) to demonstrate Proposition 1. Using the energy score proposed by Liu et al., we calculate the energy scores of the masked image features for two different masking approaches: one based on clustering the product of $e_1^i$ and $e_2^i$ ($e_1^i \cdot e_2^i, i\in\{1, \cdots d
> \}$) and the other based on clustering the absolute difference of $e_1^i$ and $e_2^i$ ($|e_1^i - e_2^i|, i\in\{1, \cdots d\}$). For both methods, we report the 0th, 25th, 50th, 75th, and 100th percentiles of the energy scores across various masking proportions. As shown in the table below, our method consistently generates higher energy scores when discriminating content $c_2$, confirming the results outlined in Proposition 1.
>
>   | Masking Proportion | Method                     | 0          | 25        | 50        | 75        | 100      |
>    | ------------------ | -------------------------- | :--------- | --------- | --------- | --------- | -------- |
>    | 5%                 | **Element product (Ours)** | **-10.78** | **-6.59** | **-5.36** | **-4.08** | **1.64** |
>    | 5%                 | Absolute difference        | -13.87     | -9.63     | -8.56     | -7.54     | -1.60    |
>    | 10%                | **Element product (Ours)** | **-9.15**  | **-5.15** | **-4.00** | **-2.77** | **2.02** |
>    | 10%                | Absolute difference        | -11.57     | -8.80     | -7.88     | -7.05     | -2.18    |
>    | 20%                | **Element product (Ours)** | **-7.46**  | **-3.46** | **-2.36** | **-1.35** | **2.66** |
>    | 20%                | Absolute difference        | -10.73     | -7.57     | -6.91     | -6.20     | -3.16    |
>    | 30%                | **Element product (Ours)** | **-6.05**  | **-2.62** | **-1.58** | **-0.63** | **2.87** |
>    | 30%                | Absolute difference        | -9.19      | -6.73     | -6.13     | -5.59     | -3.31    |
>    | 40%                | **Element product (Ours)** | **-5.71**  | **-2.23** | **-1.29** | **-0.43** | **2.70** |
>    | 40%                | Absolute difference        | -8.04      | -5.99     | -5.51     | -5.07     | -3.61    |
>    | 50%                | **Element product (Ours)** | **-5.24**  | **-1.94** | **-1.13** | **-0.37** | **2.69** |
>    | 50%                | Absolute difference        | -7.26      | -5.37     | -4.99     | -4.60     | -3.63    |
>    | 60%                | **Element product (Ours)** | **-4.93**  | **-1.72** | **-0.92** | **-0.22** | **2.72** |
>    | 60%                | Absolute difference        | -6.23      | -4.85     | -4.53     | -4.19     | -3.29    |
>    | 70%                | **Element product (Ours)** | **-3.91**  | **-1.25** | **-0.53** | **0.15**  | **2.86** |
>    | 70%                | Absolute difference        | -5.62      | -4.32     | -4.06     | -3.77     | -2.98    |
>    | 80%                | **Element product (Ours)** | **-3.11**  | **-0.77** | **-0.14** | **0.53**  | **2.20** |
>    | 80%                | Absolute difference        | -4.72      | -3.77     | -3.57     | -3.36     | -2.57    |
>    | 90%                | **Element product (Ours)** | **-2.40**  | **-0.67** | **-0.18** | **0.37**  | **2.12** |
>    | 90%                | Absolute difference        | -4.00      | -3.32     | -3.15     | -3.01     | -2.55    |
>
>
>   [1]Liu W, Wang X, Owens J, et al. Energy-based out-of-distribution detection[J]. Advances in neural information processing systems, 2020, 33: 21464-21475.
>
>  **Reply to Weakness 2:**
>
> Compared to the vanilla model, the clustering process required by our method incurs a slight increase in GPU resource usage and inference time.  We report the inference time and GPU usage when the test batch size is set to 1, as follows:
>
>    |                                   | Baseline (IP-Adapter) | Ours  |
>    | :-------------------------------- | --------------------- | ----- |
>    | GPU usage                         | 16420M                 | 16530M|
>    | Inference Time on  NVIDIA 4090 Ti | 10s                   | 11s   |
>
>
>   On the one hand, parallel computing can be employed to eliminate this additional time cost. On the other hand, instead of performing clustering to identify the masked elements, we can directly mask the top proportion (e.g., the top 5%)  of the element-wise product of $e_1$ and $e_2$.

---

> > ### Comment · Reviewer_gkxk · 2024-11-25
> >
> > Thanks for the authors' detailed response and effort. My concerns have been addressed, and I will maintain my current score.

---

> > > ### Author Response · Authors · 2024-11-25
> > > **Thanks for your valuable time and suggestions**
> > >
> > > We sincerely thank you for dedicating your valuable time to review our response. Your insightful suggestions have played a pivotal role in enhancing the overall quality of our paper. We greatly appreciate your valuable comments once again.

---

### Official Review · Reviewer_SrjP · 2024-11-03

**Soundness:** 2
**Presentation:** 2
**Contribution:** 2
**Rating:** 6
**Confidence:** 4

**Summary:**

The paper focuses on the content leakage issues in the text-to-image diffusion model for style transfer, which aims to distangle the content and style characteristics of the style-reference images for generating outputs combining text content and visual styles. It proposes a simple and training-free method to decouple the content from style in style-reference images. By masking specific content-related elements within the image features, the proposed method prevents unwanted content information from influencing the output. The proposed method was evaluated on CIFAR-10 dataset and demonstrates good results.

**Strengths:**

1. The motivation of this paper is well elaborated, and the limitations of previous methods are clearly described. Therefore, potential readers can easily understand the core problem in style transfer.
2. The structure of the paper is well-organized and the presentation is easy to follow.
3. It proposes a masking-based technique to decouple the content and style in the reference images. Fig.3 clearly demonstrates the difference between the proposed method and previous methods.

**Weaknesses:**

1. The novelty of the proposed method is limited. 1). Compared with IP-adaptor and InstantStyle, the contribution of sampling masking features in the feature space is not significant. 2). Introducing a masking mechanism is effective for manually synthesizing high-quality images, which has been demonstrated in previous studies [1-3].

2. The proposed method aims to decouple the content and style characteristics of the reference images, but this problem is not formally formulated in the paper. Therefore, it is hard to understand why the masking mechanism can achieve this goal.

3. The proposed method needs to carefully select specific features for generating desirable styles, but the selection criteria are not clearly described.

4. The paper claims that it proposes an efficient method to decouple the content and style of the reference images in the introduction part, but the paper does not show significant evidence to demonstrate its efficiency compared with the previous methods.

5. The proposed method is only evaluated on CIFAR-10 dataset, and measured with subjective metrics that are not defined clearly. Therefore, existing experiment results are insufficient to demonstrate the proposed method's advantages.

[1] Gao, Shanghua, Pan Zhou, Ming-Ming Cheng, and Shuicheng Yan. "Masked diffusion transformer is a strong image synthesizer." In Proceedings of the IEEE/CVF International Conference on Computer Vision, pp. 23164-23173. 2023.
[2] Xiao, Changming, Qi Yang, Feng Zhou, and Changshui Zhang. "From text to mask: Localizing entities using the attention of text-to-image diffusion models." Neurocomputing 610 (2024): 128437.
[3] Couairon, Guillaume, Jakob Verbeek, Holger Schwenk, and Matthieu Cord. "Diffedit: Diffusion-based semantic image editing with mask guidance." ICLR, 2023.

**Questions:**

1. It is better to elaborate on the advantages and significant contributions of the proposed method for style transfer compared to previous approaches?

2. What is the formal definition of content leakages? Additionally, how does the proposed method effectively address this issue?

3. How does the proposed method decouple the content and style characteristics of the reference images?

4. How are features selected during the style transfer process? Please provide detailed information on the selection criteria. Are the same selection criteria, including hyperparameters, applied consistently to all output images?

5. What makes the proposed method efficient? Compared to previous approaches, does it require less inference time or fewer GPU resources?
6. It would be beneficial to include more experiments on different datasets. Specifically, how does the proposed method perform when processing high-resolution images?

7. Including additional objective evaluation metrics, such as FID, LPIPS, and CLIP score, would be valuable. Since the fidelity score highly depends on the chosen classifier, how does the performance change when a different classifier is used?

---

> ### Author Response · Authors · 2024-11-23
> **Reply to weaknesses**
>
> **Reply to ``the contribution compared to IP-Adapter and InstantStyle is not significant'':**
>
> We uncover a critical yet under-explored principle: guiding with appropriately selected fewer conditions can efficiently avoid unwanted content flowing into the diffusion model,  enhancing the style transfer performances. In this paper, we introduce two strategies to appropriately select fewer conditions, i.e., the training-free masking method and the training-based Image-Adapter (as illustrated in Lines  312-317 in the paper). We demonstrate the superiority of both the training-based and training-free methods theoretically and experimentally. **Compared to the previous masking-based methods, we propose a novel marked-element selection method which masks (zero-out) the elements in the cluster with high means. This superiority of the proposed masking strategy in content removal is backed by  Proposition 1.**
>
> **Reply to `` Introducing a masking mechanism is effective has been demonstrated in previous studies'':**
>
> Although several studies have explored the effectiveness of masking mechanisms, our method differs from these approaches in several key aspects:
>
> 1. **No coupled denoising processes**: Our method avoids the need for two denoising processes, thus saving computational resources. For instance, the DIFFEDIT method requires two denoising processes—one conditioned on the query text and the other conditioned on a reference text. By contrasting the predictions of the two diffusion models, DIFFEDIT generates a mask that locates the regions needing editing to match the query text.
>
> 2. **Masking in the latent space**: Unlike DIFFEDIT, which operates on the pixel level to generate a mask highlighting the regions of the input image that need editing, our method performs masking in the latent space, bypassing pixel-level operations and patch-level manipulations.
>
> 3. **Focus on content leakage in style transfer**: While the MDT method introduces a latent masking scheme to enhance the DPMs' ability to learn contextual relations among object semantics in an image, it focuses on predicting randomly masked tokens from unmasked ones. In contrast, our method targets content leakage in style transfer. We mask feature elements that are related to unwanted content from the style reference, guided by clustering results on the element-wise product. The "From Test to Mask" method leverages the rich multi-modal knowledge embedded in diffusion models to perform segmentation. By comparing different correlation maps in the denoising U-Net, it generates the final segmentation mask.
>
>
> **Reply to  `` the problem is not formally formulated'':**
>
> We apologize for the confusion. We provide the formulation of the problem for clarity:
>
> In the context of style transfer, given a style reference $c_1$, the content of the style reference $c_2$, and the target text prompt $c_3$,
> the text-driven style transfer aims to generate a plausible target image by combining the content of the target text prompt with the style of the style reference, while ensuring that the unwanted content from the style reference does not transfer into the generated result.
>
> **Reply to  `` the selection criteria are not clearly described'':**
>
> We apologize for the confusion. We introduced the selection criteria in Lines 212–231 in Section 3.1,  as illustrated in Figure 3(a).
>
>  **Reply to  ``The paper does not show significant evidence to demonstrate its efficiency compared with the previous methods'':**
>
> We provide visual comparisons in Figures 6, 7, 10, and 11 to demonstrate the superiority of our method compared to previous approaches. In comparison to the baseline models, IP-Adapter and InstantStyle, we present generation results across various coefficient values in Figure 11 of Appendix A.5. These results highlight that both IP-Adapter and InstantStyle heavily rely on test-time coefficient tuning for style strength, requiring users to engage in a labour-intensive process to achieve a balanced synthesis between target content and style. Particularly in high-coefficient scenarios, both models experience significant content leakage from the style references and a loss of text fidelity. In contrast, our method produces satisfactory results even in high-coefficient settings.
>
>  **Reply to  ``The proposed method is only evaluated on CIFAR-10 dataset'':**
>
> We apologize for the confusion.  **The proposed method was not only evaluated on our newly constructed dataset based on the classes of CIFAR-10 (not a real CIFAR-10 dataset) but also on the benchmark dataset Stylebench, proposed in StyleShot, both on text-driven and image-driven style transfer.**

---

> ### Author Response · Authors · 2024-11-23
> **Reply to questions**
>
> **Reply to Q1:**
>
> Thank you for your helpful suggestion! We will add a part to highlight the comparison between our method and the previous methods. For details about the comparison, please refer to the reply to weaknesses.
>
> **Reply to Q2:**
>
> Thank you for your helpful suggestion. We will provide a formal definition of the problem in the revised paper. As for how our method addresses this issue, we have detailed our approach in Lines 212-231. The effectiveness of our method is supported by the highest energy of  $\mathcal{E}(c_2, x_t)$, achieved through the proposed masked element selection criteria, as shown in Proposition 1. This effectively reduces the likelihood of content in style reference, leading to superior performance in content removal. We also provide the simulation result in the reply to Reviewer gkxk, confirming the results in Proposition 1.
>
> **Reply to Q3:**
>
> We detailed the proposed method in Section 3.1. The proposed masking strategy to decouple the content and style characteristics was illustrated in Lines 212-231.
>
> **Reply to Q4:**
>
> The proposed masking strategy is detailed in Section 3.1. For the proposed masking-based method, we use the same selection criteria based on the K-means cluster and set the clustering number as 2.
>
> **Reply to Q5:**
>
> Compared to training-based methods, such as InST, the proposed masking-based method does not require a training process and instead manipulates the image condition of the IP-Adapter in a plug-and-play manner. By simply performing clustering on the element-wise product of $e_1 \cdot e_2$ based on IP-Adapter, our method effectively mitigates the content leakage issue. While the clustering process introduces a slight increase in GPU resource usage and inference time compared to the vanilla IP-Adapter model, it offers an efficient training-free solution for content removal.
>
>
>    |                                   | Baseline (IP-Adapter) | Ours  |
>    | :-------------------------------- | --------------------- | ----- |
>    | GPU usage                         | 16420M                 | 16530M |
>    | Inference Time on  NVIDIA 4090 Ti | 10s                   | 11s   |
>
>
>  **Reply to Q6:**
>
> We perform experiments on our constructed dataset  (**not a real CIFAR-10 dataset**), which uses content objects of CIFAR-10 but with various styles. The images are generated using the MACE code, and each image has a resolution of 512x512. All relevant details have been discussed in Lines 352-358 in the main paper. More importantly, we also evaluate our method on the benchmark dataset StyleBench and report the corresponding results in Figure 6 and Figure 12.
>
>  **Reply to Q7:**
>
> The evaluation metrics, **FID** and **LPIPS**, are primarily used to assess the quality of generated images by measuring the similarity between real and generated images. However, a high similarity does not necessarily indicate low content leakage or high style similarity. Following your insightful suggestion, we report the image alignment and text alignment scores for our method and its counterpart, InstantStyle, using different CLIP classifiers, based on the StyleBench dataset. The results, shown in the table below, reveal that while InstantStyle achieves slightly higher text alignment, it significantly sacrifices image alignment with the style reference across various CLIP classifiers. As illustrated in Figure 6 and Figure 12, InstantStyle avoids content leakage at the cost of style degradation. By leveraging appropriately fewer conditions, our method achieves a better balance between target content and style, producing more effective style transfer results.
>
>
>    |                 | ViT-B/32 |              | ViT-L/14 |              | ViT-H/14 |
>    | --------------- | -------- | ------------ | -------- | ------------ | ----- |
>    | Image Alignment |          |              |          |              |       |
>    | InstantStyle    | 0.575    | InstantStyle | 0.579    | InstantStyle | 0.352 |
>    | ours            | 0.657    | ours         | 0.608    | ours         | 0.403 |
>    | Text Alignment  |          |              |          |              |       |
>    | InstantStyle    | 0.275    | InstantStyle | 0.218    | InstantStyle | 0.263 |
>    | ours            | 0.265    | ours         | 0.212    | ours         | 0.258 |

---

> > ### Comment · Reviewer_SrjP · 2024-11-26
> >
> > Thank you for the detailed responses, which have addressed my concerns.  I would like to raise my scores after reading the responses and the revised version.

---

> > > ### Author Response · Authors · 2024-11-26
> > > **Thanks for your valuable time and suggestions**
> > >
> > > Thank you for your recognition and support of our work. We sincerely appreciate the time and effort you have devoted to considering our response. Your constructive suggestions have greatly enhanced the quality of our manuscript.

---

### Official Review · Reviewer_eaaS · 2024-11-05

**Soundness:** 3
**Presentation:** 2
**Contribution:** 3
**Rating:** 6
**Confidence:** 5

**Summary:**

This paper presents a style transfer method that can preserve style in the style reference image while ignored the content injection for the final style transfer result.  The key idea of the proposed method is based on IP-adpator while masked out the some image tokens from the reference image.  The masking strategy is first clustering the product feature of style image and content image and then filtering out the feature tokens of high means in style features.   To approve the masked strategy is effective, the authors also provided several theoretical justifications.

**Strengths:**

The key strengths in this paper are the insights of analyzing different style transfer methods including IP-adpator and Instant Style. With those observations, the proposed method provide a masking solution to demonstrate that removing certain tokens especially the token with high correlations between content and style will result a high fidelity stylized images.

**Weaknesses:**

There are several weaknesses in this paper:

1. The masked image feature is questionable. Although the proof demonstrate the divergency in a theoretical way, it is clear that it only demonstrate the divergence by comparing with InstantStyle rather than the method itself.   Image token selection is based on the product between content and style feature, such computation is more likely filtering about the foreground style feature.  Thus why not only encode the background  or non-object related style patches?

2. The visual comparison is not in a fair comparison.  Lots of the results are in a cherry pick manner.  For example, StyledDrop and StyleShot focus more in tradition painting like style transfer, while the experiments showing more like photo as a reference image, which not makes much sense.  Moreover, the Figure 8 compare the results with InstantSyle but not StyleShot is also not fair since the setting is more like the traditional style transfer.

**Questions:**

Please double check the experiments. For example, I screenshot one style reference image and use the official styleshot demo, I could generate better results shown in the paper.

Another fair comparison is leveraging existing benchmark (the images used in styledrop and styleshot) and shown more proposed results on that.

There are also some questions on the visual results. For example, the Figure 1 shows that the proposed method also could not generate visual plausible results especially preserving the style in style reference image. In Figure 8, it is clear to see that some cases InstantStyle gives better results.  As it preserves the content while generates less artifacts.  Thus more results on the existing benchmark maybe a good justification.

**Details Of Ethics Concerns:**

The paper clearly showing some weaknesses but lack of such discussion,  especially on human related stylization.

---

> ### Author Response · Authors · 2024-11-23
> **Reply to weaknesses**
>
> **Reply to ``Demonstrate the divergence by comparing with InstantStyle rather than the method itself'':**
>
> In Theorem 1, we compare the proposed method, which masks certain elements of the image condition, with the vanilla method. It should be noted that the vanilla method, without the masking operation, uses conditions $c_1$, $c_2$, and $c_3$, forming a family of models that includes the InstantStyle method.
>
> **Reply to ``Why not only encode the background or non-object related style patches'':**
>
> It is true that we can encode the background or non-object-related style patches into the image encoder to prevent content leakage. To achieve this, we can use the GroundingDINO [1] and SAM [2] models to locate the non-object region in the image. However, compared to the proposed method, which performs masking in the latent space, masking patches in the image requires more computational resources and inference time.
>
> **Reply to `` the experiments showing more like photo as a reference image'':**
>
> In this work, we focus on the issue of content leakage in style transfer, an important challenge that has been explored by several recent studies [3,4]. We have included a comparison between our method and these studies [3,4], along with visual comparisons available at the following links: https://drive.google.com/file/d/1jjAgViV9Z7MTrES0eMhaBPlpS2ZtyEVE/view?usp=sharing and https://drive.google.com/file/d/1zQQa2XHlTBspOJ1IvC6VKQnRxEuu2Lvc/view?usp=drive_link. A detailed discussion can be found in Reply 1 to Reviewer nfUg, and the related points have been incorporated into the revised manuscript.
>
> Here are further explanations of the comparison in this work:
>
> 1) To fully demonstrate the effectiveness of the proposed method in mitigating the leakage of various content from the image reference into the generated image, we create style references by combining 21 different styles and objects (from CIFAR-10). We analyze the experiment results in Section 4.1.
>
> 2) In Section 4.2, we also conducted experiments on the standard benchmark dataset, StyleBench, proposed in StyleShot, which includes 73 different styles, comprising both non-object and object style references. As shown in the 5th and 6th rows of Figure 6, **when using non-object style references, both StyleDrop and StyleShot suffer from style degradation or loss of text fidelity.**  **We provide additional results to compare the performance of our method with previous methods using non-object style references** at https://drive.google.com/file/d/1ZBISv9HsiSyw5yNoHPOdHamN8kqapoxG/view?usp=sharing
>
> **Reply to `` Figure 8 compare the results with InstantSyle but not StyleShot'':**
>
> We provided visual comparisons between StyleShot and our method for image-driven style transfer in Figure 7. In Figure 8, we compare the proposed masking method with the feature subtraction approach of InstantStyle, **using the same model configuration, including the StyleShot style encoder**, random seed, guidance scale, and other parameters.
>
> [1] Liu S, Zeng Z, Ren T, et al. Grounding dino: Marrying dino with grounded pre-training for open-set object detection[J]. arXiv preprint arXiv:2303.05499, 2023.
>
> [2] Ravi N, Gabeur V, Hu Y T, et al. Sam 2: Segment anything in images and videos[J]. arXiv preprint arXiv:2408.00714, 2024.
>
> [3] Rout L, Chen Y, Ruiz N, et al. RB-Modulation: Training-Free Personalization of Diffusion Models using Stochastic Optimal Control[J]. arXiv preprint arXiv:2405.17401, 2024.
>
> [4] Xing P, Wang H, Sun Y, et al. Csgo: Content-style composition in text-to-image generation[J]. arXiv preprint arXiv:2408.16766, 2024.

---

> ### Author Response · Authors · 2024-11-23
> **Reply to questions**
>
> **Reply to ``StyleShot's better results than those shown in the paper'':**
>
> Thanks for your suggestion!
> It is likely that the image generation results of the diffusion model exhibit high randomness. To account for this, we compare the style transfer results by reporting the average results of four generations for each combination of style reference and target text prompt.
>
> **We also provide multi-sample generalization results for each combination of style reference and target text prompt at** https://drive.google.com/file/d/1XUCFhPFsrgD49uuQosNrU323zWxAqbbX/view?usp=drive_link. Content leakage and loss of text fidelity are marked for the one-to-one image generation results in Figure 20 of the revised manuscript. **To avoid the influence of randomness, we ensure that all model configurations remain consistent,** including the random seed, guidance seed, denoising steps, and other parameters. From the one-to-one comparison, we observe that our method significantly reduces content leakage and alleviates loss of text fidelity, consistently refining StyleShot’s results across all combinations.
>
> **Reply to `` Comparison with existing benchmark'':**
>
> In Section 4.2, we conducted experiments on the existing benchmark (the StyleBench dataset used in StyleShot) to demonstrate the effectiveness of our method. Visual comparisons were provided in Figure 8 and Figure 12 of the paper.
>
> **Reply to ``Questions on the visual results'':**
>
> The visual results in Figure 1 are based on the StyleBench dataset from StyleShot. Additional results are provided in Figure 6 and Figure 15 in the paper. Regarding the results in Figure 1, our method effectively avoids content leakage while exhibiting less style degradation compared to the previous method.
> In Figure 8(a), InstantStyle’s results tend to disrupt the style information extracted by StyleShot’s style encoder due to image-text misalignment. In Figure 8(b), InstantStyle also experiences style disruption, while our method alleviates this issue by using fewer appropriately-selected conditions.

---

> > ### Comment · Reviewer_eaaS · 2024-11-26
> >
> > Thanks for providing the new results on the traditional style transfer examples.  The new results look promising and I suggest to add it back to the final draft. They are actually better than the figures showing in the paper. Most of my concerns have been addressed.

---

> > > ### Comment · Reviewer_eaaS · 2024-11-26
> > >
> > > A follow up question is the human related generation always have strong distortions.  It may have some concerns for ethics.  Please discuss the risk especially sharing more examples of text prompts contain different gender, skin color , ages and etc,. Otherwise, it will forbid for better ratings.

---

> > > > ### Author Response · Authors · 2024-11-26
> > > >
> > > > We sincerely thank the reviewer for reminding us of the ethical issues. This work aims to make a positive impact on the field of AI-driven image generation. We aim to facilitate the creation of images with diverse styles, but we expect all related processes to comply with local laws and be used responsibly.
> > > >
> > > > The use of AI to generate human-related images, particularly those involving characteristics such as skin color, gender, age, and other demographic factors, raises complex ethical questions. We are aware that the generation of images involving these attributes must be handled with care to avoid reinforcing stereotypes, perpetuating discriminations, or contributing to the misrepresentations of certain groups. We take these concerns very seriously and believe that AI should be used in a way that promotes fairness, inclusion, and respect for all individuals. Here, we give several examples of text prompts containing different genders, skin colors, and ages, as shown in Figure 22 in Appendix A.10.
> > > > We observe that in most cases, our method is able to generate images with diversity. However, there are certain cases that general image generation methods can be misused.
> > > >
> > > > In light of these considerations, we have added an ethics statement in Appendix A.10 including the term that the code and methodology in this paper shall be used responsibly. Users are expected to utilize this material in a way that avoids any potential bias related to sensitive attributes such as gender, race, age, and other demographic factors. We believe that the responsible use of AI-driven image generation tools is essential to fostering ethical and equitable outcomes in the field.

---

> > > ### Author Response · Authors · 2024-11-26
> > >
> > > We sincerely appreciate your valuable comments and the time you have dedicated to considering our response. We have revised the visual comparison in Figure 1 and included additional traditional style transfer examples in Appendix A.9 in the manuscript.

---

### Comment · Area_Chair_jRrp · 2024-11-25
**Please check the authors' responses**

Dear reviewers,

Could you please check the authors' responses, and post your message for discussion or changed scores?

best,

AC

---

### Meta-Review · Area_Chair_jRrp · 2024-12-19

**Metareview:**

This paper tackles the style transfer problem aiming at text-to-image generation of stylized image by transferring the style from reference image. The fundamental contribution of this work is the proposed training-free masking-based method that decouples content from style, by simply masking specific style reference’s features preventing the content leakage from the stylized reference image. The masking is based on the clustering of element-wise product between image and text features and discard elements in the high-means cluster. The experiments validated the effectiveness of this proposed approach for style transfer.

This work received generally positive comments from the reviewers after rebuttal, and the final scores are 6, 6, 6, 8. The major strength of this work is on the training-free masking strategy. Considering the overall positive final recommendations of reviewers, the paper can be accepted, but requiring to incorporate the revisions into the final version.

**Additional Comments On Reviewer Discussion:**

Reviewer eaaS raised questions on the masking technique of image features and fair comparisons with InstantSyle and StyleShot. The authors provided link to more comparison results, and addressed the reviewer's concerns. Reviewer SrjP questioned on the limited novelty, reason of masking technique, and insufficient results.  Reviewer gkxk asked the reason why clustering the element-wise product of e1 and e2, and the inference speed. Reviewer nfUg gave a score of 8, and suggested on the comparison with more recent baselines, and asked questions on some details. The authors' rebuttal well solved these concerns, but the authors are suggested to include these suggestions/revisions in the final version.

---

### Decision · Program_Chairs · 2025-01-22

Accept (Poster)